# Explaining How Visual, Textual and Multimodal Encoders Share Concepts

## Abstract

Sparse autoencoders (SAEs) have emerged as a powerful technique for extracting human-interpretable features from neural networks activations. Previous works compared different models based on SAE-derived features but those comparisons have been restricted to models within the same modality. We propose a novel indicator allowing quantitative comparison of models across SAE features, and use it to conduct a comparative study of visual, textual and multimodal encoders. We also propose to quantify the *Comparative Sharedness* of individual features between different classes of models. With these two new tools, we conduct several studies on 21 encoders of the three types, with two significantly different sizes, and considering generalist and domain specific datasets. The results allow to revisit previous studies at the light of encoders trained in a multimodal context and to quantify to which extent all these models share some representations or features. They also suggest that visual features that are specific to VLMs among vision encoders are shared with text encoders, highlighting the impact of text pretraining.

## 1 Introduction

Sparse autoencoders offer promising insights for concept-based analysis of neural networks (Bricken et al., 2023; Cunningham et al., 2023). By learning sparse representations of model activations, SAE allow the extraction of interpretable features from both language and vision models. Recent works compare different models upon SAE features, by constructing a common concept space (Thasarathan et al., 2025), or by quantifying similarities between models (Wang et al., 2025). However, these studies are constrained in scope: they typically focus on a small number of models (two to three) and limit comparisons to a single modality.

In contrast, this paper introduces a large-scale comparative study of internal representations across 21 encoder models. Our contribution departs from previous efforts in two key aspects. First, the scale of our analysis substantially exceeds earlier studies, which have been restricted to pairwise or small-group comparisons using a single dataset (Thasarathan et al., 2025; Wang et al., 2025). Second, we explicitly address multimodal comparison, using textual, visual, and multimodal encoders of varying sizes (Sec 3), without being bound to models comprising both image and text encoders.Furthermore, we also use three datasets of text-image pairs as input, in particular to study the effect of specific domains in the context of SAE-based interpretability. To provide a finer analysis of this large study, we introduce two new tools (Sec 2). The *wMPPC* (*weighted Maximum Pairwise Pearson Correlation*) is a similarity indicator between models, which is formally the expectation of the per-feature maximal correlation under sampling by activation mass. The *Comparative Sharedness* of individual features allows the identification of features from a given model that are better shared with a class of model than another. The main remarkable outcomes of this study are: (i) The shared information between models of different modalities is to be found mostly in the last layer of each model (Sec 3.2) (ii) *wMPPC* reveals differences in image-text alignment quality between datasets (Sec 3.3). (iii) We establish a typology of SAE features learnt on CLIP visual encoder that are shared with multiple VLMs, better than with classical visual foundation models (Sec 3.4). Such features are related to high-level semantic concepts, such as specific geographical regions, or even purely textual information. (iv) We find this typology to be similar to the one obtained while looking for visual features of CLIP that are better shared with text encoders (using image captions) than with visual foundation models (Sec 3.5). Therefore, we highlight the impact of text pretraining on image understanding, by isolating individual concepts that are specific to those models.

## 2 METHOD

We aim to quantify the similarity of interpretable features from two large language models (LLM encoders) $A$ and $B$. For input data $x \in \mathcal{D}$, the interpretable features (resp. $f_i^A$ and $f_j^B$) are identified with Sparse Autoencoders (SAEs) trained on the model activations. Each SAE consists of two linear layers with rectified linear units (ReLU) and aims at minimizing an error reconstruction on the input activations. We use TopK sparse autoencoders (Gao et al., 2025; Makhzani & Frey, 2013), that directly constrain sparsity via an activation function, by only keeping the $k$ highest activations and setting others to zero. The SAE is finally trained with a mean square error loss using all patches (images) or token (text) of the input $x$. However, at inference, we only consider the features corresponding to the global representation of data samples (e.g. CLS token), in order to compare the SAE features of models with different patch sizes, tokenizers, or even different modalities. In the following, we informally introduce two new tools to compare $A$ and $B$, with details and proofs in Appendix B.

### 2.1 WEIGHTED MPPC

In order to compare models $A$ and $B$ upon their SAE features, we extend the MPPC indicator (Wang et al., 2025). For each of the $n$ interpretable features $f_i^A$ of model $A$ and each feature $j$ of model $B$, the Pearson correlation is $\rho_{ij} = corr(f_i^A, f_j^B)$. The maximum pairwise Pearson correlation of feature $i$ (MPPC per-feature) is $\rho_i^{A \to B} = \max_j \rho_{ij} \in [-1, 1]$ and the MPPC to compare $A$ to $B$ is:

$$\text{MPPC}^{A \to B} = \frac{1}{n} \sum_{i=1}^{n} \rho_i^{A \to B} \tag{1}$$

We introduce nonnegative weights $w_i \geq 0$ with $\sum_{i=1}^{n} w_i = 1$ and define the weighted MPPC as:

$$\text{wMPPC}^{A \to B} = \sum_{i=1}^{n} w_i \rho_i^{A \to B} \tag{2}$$

Since $\text{wMPPC}^{A \to B}$ is a convex combination of the $\rho_i^{A \to B}$ we have $\min_i \rho_i^{A \to B} \leq \text{wMPPC}^{A \to B} \leq \max_i \rho_i^{A \to B}$ thus $\text{wMPPC}^{A \to B}$ cannot produce values outside the range of the per-feature correlations; it simply shifts emphasis. Let us consider $a_i(x) \in \mathbb{R}$ the activation (nonnegative scalar because of the ReLU) of the SAE feature $f_i^A$ on input $x \in \mathcal{D}$, and the cumulative activation of feature $i$ over $\mathcal{D}$ noted as $S_i^A = \sum_{x \in \mathcal{D}} a_i(x)$.

**Proposition 1.** *If one considers the normalized weights $w_i = \frac{S_i^A}{\sum_{\ell=1}^{n} S_\ell^A}$ we have:*

$$\text{wMPPC}^{A \to B} = \frac{1}{\sum_{\ell=1}^{n} S_\ell^A} \sum_{x \in \mathcal{D}} \sum_{i=1}^{n} a_i(x) \rho_i^{A \to B} \tag{3}$$

*Therefore $\text{wMPPC}^{A \to B} = \mathbb{E}_g \left[ \rho_i^{A \to B} \right]$ where $g$ is the joint distribution that samples a datapoint $x$ in $\mathcal{D}$ and then samples feature $i$ with probability proportional to $a_i(x)$.*

Hence, with weights reflecting how important a feature is on the dataset $\mathcal{D}$, $\text{wMPPC}^{A \to B}$ *is the expectation of the per-feature maximal correlation under sampling by activation mass*: it measures the similarity between $A$ and $B$ "as experienced by the data" rather than treating all features equally, emphasing features that contribute most and decaying those that are dormant or negligible on $\mathcal{D}$. If most downstream decisions (or most of the model's computational mass) depend on high-activation features, it measures similarity "where it matters".

Since we use TopK-SAE, let us consider the Top-$k$ cumulative activation $c_i^{(k)} = \sum_{t=1}^{k} a_i^{(t)}$ with $a_i^{(1)} \geq a_i^{(2)} \geq \cdots \geq a_i^{(k)}$, and denote $C^{(k)} = \sum_{i=1}^{n} c_i^{(k)}$. The Top-$k$ activation-weighted MPPC is:

$$\text{wMPPC}_{(k)}^{A \to B} = \frac{1}{C^{(k)}} \sum_{i=1}^{n} c_i^{(k)} \rho_i^{A \to B} \tag{4}$$

One can derive an incremental update formula for $\text{wMPPC}_{(k)}^{A \to B}$ (see Proposition 2 in Appendix B), which gives a practical criterion to choose the sparsity level of the Top-k SAE (with $a_i^{(k)} > 0$):

**Corollary 1.** *(monotonicity condition) A sufficient condition for* $\text{wMPPC}^{A \to B}_{(k+1)} \geq \text{wMPPC}^{A \to B}_{(k)}$ *is:*

$$\sum_{i=1}^{n} a_i^{(k+1)} \big( \rho_i^{A \to B} - \text{MPPC}_w^{(k)} \big) \geq 0 \tag{5}$$

thus when one relaxes sparsity (increases $k$), it improves the wMPPC if the additional activations are concentrated on features that are similar to model $B$ with $\rho_i^{A \to B} \geq \text{MPPC}_w^{(k)}$.

## 2.2 COMPARATIVE SHAREDNESS TO IDENTIFY INDIVIDUAL FEATURES OF A MODEL

Using *wMPPC*, we are able to evaluate whether two models share the same features on average, at a global scale. But a concept-level analysis can give even more insight on the inner representations of models, by establishing a typology of concepts that are specific to a group of models, but not shared with another. For a given feature $\boldsymbol{f}_i$ from a model $M$, we define the *Comparative Sharedness* $\Delta_i^{M \to A,B}$ by:

$$\Delta_i^{M \to A,B} = S_i^M \times (\rho_i^{M \to A} - \rho_i^{M \to B})(\rho_i^{M \to A} + \rho_i^{M \to B}) \tag{6}$$

where $S_i^M = \sum_{\boldsymbol{x} \in \mathcal{D}} \boldsymbol{f}_i(\boldsymbol{x})$ is computed over all the input examples to weight the importance of the feature. Hence, $\Delta_i^{M \to A,B}$ is the difference of *wMPPC* contributions of the considered feature $\boldsymbol{f}_i$ of a model $M$ towards $A$ and $B$ (measuring if the feature is "better shared" with $A$ than with $B$), multiplied by the sum of their maximum correlations, in order to favour features that have high correlations with at least one model. This way, the features of $M$ with a high value of $\Delta_i^{M \to A,B}$ are "well shared" with $A$, but not with $B$. Comparative Sharedness without the sum of $\rho_i$ term has been experimented, but in practice led to less diversity, as it was heavivly dominated by $\rho^{M \to A}$.

In a cross-modal context, it is for instance interesting to identify the features of a visual encoder $M$ which are highly correlated to the textual features of a model $A$ but not to those of another visual encoder $B$. It then exhibits a specificity of $M$ with regards to $B$. The approach is even more interesting when it is extended to two groups of models $\boldsymbol{G}$ and $\boldsymbol{H}$, since it can exhibit the features that some encoder handles in contrast to some other encoders. To find features shared with every model in $\boldsymbol{G}$, but with no model in $\boldsymbol{H}$, we define the *Generalized Comparative Sharedness*:

$$\Delta_i^{M \to \boldsymbol{G},\boldsymbol{H}} = S_i^M \times \left( \big( \min_{G_i \in \boldsymbol{G}} \rho_i^{M \to G_i} \big)^2 - \big( \max_{H_i \in \boldsymbol{H}} \rho_i^{M \to H_i} \big)^2 \right) \tag{7}$$

In the vein of the example above, if we consider a visual encoder $M$, a set $\boldsymbol{T}$ of several textual encoders and a set $\boldsymbol{V}$ of several visual encoders, high values of $\Delta_i^{M \to \boldsymbol{T},\boldsymbol{V}}$ would be associated with the features of $M$ that are specifically correlated to some textual features (among a large set) while being different from other visual features.

## 2.3 ON COMPUTATIONAL TRACTABILITY

Computing *MPPC*$^{A \to B}$ or *wMPPC*$^{A \to B}$ requires computing $n_A \times n_B$ Pearson correlations, with $n_A$ and $n_B$ the number of SAE features learnt on models $A$ and $B$. In practice, computing *wMPPC*$^{A \to B}$ on every layer of models requires tens of billions of correlations, between vectors as long as the dataset. The Pearson correlation $r(X,Y) = \frac{\mathbb{E}[(X - \mu_X)(Y - \mu_Y)]}{\sigma_X \sigma_Y}$ is computed from N samples of the random variables $X$ and $Y$. With $\tilde{X} = \frac{X - \mu_X}{\sigma_X}, \tilde{Y} = \frac{Y - \mu_Y}{\sigma_Y}$, we have $r(X,Y) = \frac{\tilde{X} \cdot \tilde{Y}}{N}$. Therefore, the Pearson correlation can be seen as a dot product between standardized vectors. All the correlations required by *wMPPC* can therefore be computed in a single matrix multiplication, between the matrices of standardized features. Block matrix multiplication (or chunking) can be used to solve potential memory issues. For example, computing *wMPPC* on COCO between two models with 24 layers $\times$ 8192 features (largest models of this study, see Appendix A) requires $9.14 \times 10^{15}$ FLOPs, and 469s on a single Nvidia A100 GPU, using FP32 precision at peak theoretical throughput. In practice, 5 runs of this settings took $608.6 \pm 5.9$ seconds on a computer cluster. Considering only the last layer of each model (like required to compute *Comparative Sharedness*) divides the number of operations by the number of layers of each model. Using the same two models, it would require $1.59 \times 10^{13}$ FLOPs, 0.81s at peak throughput on a single A100 GPU, and $1.01 \pm 0.01$ seconds on 2880 timed runs.

Table 1: $wMPPC^{source \rightarrow target}$ (all layers) on COCO, for 6 large image and text encoders

| Target
Source | | Image | | | Text | | |
|---|---|---|---|---|---|---|---|
| | | CLIP (I) | SigLIP2 (I) | DinoV2 | CLIP (T) | SigLIP2 (T) | BERT |
| Image | CLIP (I) | 1 | 0.446 | 0.486 | 0.209 | 0.131 | 0.194 |
| | SigLIP2 (I) | 0.514 | 1 | 0.509 | 0.272 | 0.171 | 0.251 |
| | DinoV2 | 0.556 | 0.518 | 1 | 0.250 | 0.153 | 0.233 |
| Text | CLIP(T) | 0.253 | 0.275 | 0.246 | 1 | 0.351 | 0.428 |
| | SigLIP2 (T) | 0.045 | 0.050 | 0.043 | 0.256 | 1 | 0.578 |
| | BERT | 0.182 | 0.194 | 0.177 | 0.346 | 0.287 | 1 |

## 3 EXPERIMENTS

### 3.1 EXPERIMENTAL SETUP

**Models** We consider several classes of models, with different architectures and various sizes. For VLMs, we use CLIP (Radford et al., 2021), DFN (Fang et al., 2024) and SigLIP2 (Tschannen et al., 2025), each having a visual and a textual encoder; for language models, we consider BERT (Devlin et al., 2019) and DeBERTa (He et al., 2021); for visual foundation models (FM), we use DinoV2 (Oquab et al., 2023) and ViT (Dosovitskiy et al., 2020). We also consider MambaVision (Hatamizadeh & Kautz, 2024) as a visual FM, but its architecture is different from a succession of transformer blocks, with blocks comprising both Mamba mixers and self-attention. Therefore, we consider it only at the last layer, as the choice of the network stages to consider as "layers" could cause drastic and arbitrary modifications towards *wMPPC* at a model-level. All these models were tested in different sizes, using the *base* and *large* models. More details on these models can be found in appendix A.

**Datasets** We consider two general domain datasets: COCO (Lin et al., 2014), in particular the `train2017` split with 118 287 images and corresponding captions, and a subset of 61 642 image-text pairs from Laion-2B[1] (Schuhmann et al., 2022). We also consider a dataset in a specific domain, with images and captions: Oxford-102 Flowers[2] (Nilsback & Zisserman, 2008), with 8 189 image-text pairs.

**Implementation details** The datasets are used as input of the encoders and we use the activations of their layers as training data of the sparse autoencoders. SAEs are thus learnt in the residual stream after each transformer block, for every model. SAEs are trained with the Adam optimizer, using $\beta_1 = 0.9$ and $\beta_2 = 0.999$. The learning rate is set to $5 \cdot 10^{-5}$ for all configurations. Also, we initialize $W_{enc}$ as $W_{dec}^T$ as per (Gao et al., 2024), in order to prevent "dead latents" (never activated features). Our SAEs use a TopK architecture, with $k = 32$, meaning that training is achieved by using 32 sparse codes to represent every input. This value was chosen as it was the smallest power of 2 obtaining no dead latents on COCO with CLIP-ViT-L/14. Finally, we use an expansion factor of 8, similarly to (Thasarathan et al., 2025), which means that the intermediate representation of SAEs is 8 times as large as their input dimension. All the SAEs of this study were trained using the same SAE hyperparameters, in order to perform a systematic analysis of their learnt features.

### 3.2 COMPARISON AT THE MODEL LEVEL

We compute *wMPPC* between the image (I) and text (T) encoder of CLIP and SigLIP2, Dino v2 and BERT, using the large (L-size) version of each and COCO as input dataset. When we consider all the layers of these 6 encoders, the results are reported in Table 1. At a model level, comparisons between encoders with the same modality obtain much higher *wMPPC* than cross modality comparisons, even when considering the two encoders of a same VLM. SigLIP2 reaches state-of-the-art performance on various vision-language tasks (Tschannen et al., 2025). Both *wMPPC* between its

---

[1]From the dataset https://huggingface.co/datasets/MayIBorn/laion_2b_en_subset_70666, we collected the images that were still available at the given urls, resulting in the 61 642 image-text pairs.

[2]https://huggingface.co/datasets/efekankavalci/flowers102-captions

Table 2: $wMPPC^{source \to target}$ (last layers) on COCO, for 6 large image and text encoders

| Source \ Target | | Image | | | Text | | |
| --- | --- | CLIP (I) | SigLIP2 (I) | DinoV2 | CLIP(T) | SigLIP2 (T) | BERT |
| Image | CLIP (I) | 1 | 0.278 | 0.208 | 0.220 | 0.128 | 0.203 |
| | SigLIP2 (I) | 0.320 | 1 | 0.236 | 0.274 | 0.153 | 0.249 |
| | DinoV2 | 0.270 | 0.290 | 1 | 0.254 | 0.142 | 0.216 |
| Text | CLIP(T) | 0.255 | 0.284 | 0.211 | 1 | 0.192 | 0.286 |
| | SigLIP2 (T) | 0.054 | 0.062 | 0.042 | 0.134 | 1 | 0.297 |
| | BERT | 0.183 | 0.195 | 0.136 | 0.237 | 0.172 | 1 |

two encoders (0.050 and 0.171) are nevertheless lower than *wMPPC* between BERT and DinoV2 (0.177 and 0.233).

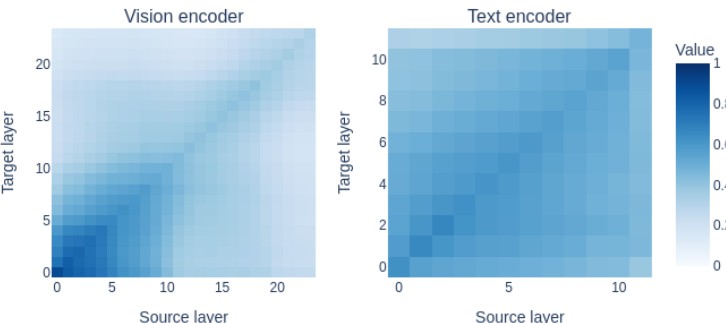

Figure 1: Layerwise *wMPPC* between 2 SAEs trained on each encoder of CLIP

However, nothing guarantees that the early layers of image and text encoders would correspond to features of the same semantic level. Therefore, we train two SAEs on each encoder of a CLIP-ViT-L/14 model. We then represent the layerwise *wMPPC* of the two encoders in Figure 1. The SAEs learnt on the text encoder obtain similar *wMPPC* for every pair of layers considered. For the image encoder, we encounter much higher *wMPPC* on early layers, and *wMPPC* stays concentrated along the diagonal, with layers of different levels obtaining low *wMPPC*, hence representing very different features. Therefore, as suggested by previous works (FEL et al., 2023), features with the highest semantic level should be found in the last layer of vision encoders.

We compute *wMPPC* considering only the last layer of each model, with the results being reported in Table 2. In that case, *wMPPC* decreases substantially for same-modality comparisons (even for two text encoders), but stays stable or even increases for cross-modal comparisons. Therefore, we deduce that the shared information between models of different modalities is to be found mostly in the last layer of each model.

Furthermore, we conduct a similar analysis with four additional encoders, namely the image and text encoder of DFN, a ViT (image encoder) and DeBERTa (text encoder), resulting in a total of 10 large image and text encoders. In Table 3, we report the average of $wMPPC^{A \to B}$ with each encoder modality for $A$ and $B$. "Image $\to$ Text" represents the average *wMPPC* with any image encoder as the source and a text encoder as the target (respectively for other combinations). Same-encoder comparisons are omitted from the average.

Using smaller versions of the same encoders (B-size instead of L-size) on COCO, *wMPPC* appears to be very similar (or slightly higher) as shown in Table 3, thus leading to the same conclusions as using L-size models.

Table 3: Average *wMPPC* for all model pairs, combined by the modality of the source and target encoders, and tested on different datasets and for different model sizes. Each number correspond to the average score of a quarter of table such as Table 1 (all) or Table 2 (last), without the '1' on the diagonal and all models of each type (instead of 3 only in Table 1 and Table 2), either in their large (L with 10 encoders) or basic (B with 10 other encoders) size. Detailed results for each model are provided in Appendix D.

| Layers | Input dataset | Models size | Image $\rightarrow$ Image | Image $\rightarrow$ Text | Text $\rightarrow$ Image | Text $\rightarrow$ Text |
|--------|---------------|-------------|-----------|-----------|-----------|-----------|
| All | COCO | L | 0.463 | 0.204 | 0.168 | 0.367 |
| | COCO | B | 0.509 | 0.225 | 0.178 | 0.405 |
| | Laion | L | 0.470 | 0.146 | 0.140 | 0.524 |
| | Flowers | L | 0.548 | 0.180 | 0.129 | 0.447 |
| Last | COCO | L | 0.265 | 0.213 | 0.173 | 0.249 |
| | COCO | B | 0.281 | 0.222 | 0.176 | 0.275 |
| | Laion | L | 0.187 | 0.119 | 0.122 | 0.308 |
| | Flowers | L | 0.377 | 0.133 | 0.166 | 0.281 |

### 3.3 ALTERNATIVE INPUT DATASET

Previous studies dealing with SAE-based interpretability relied on a single input dataset to generate the activations on which the SAE are learnt. We extend the scale of the experiment by using two additional datasets to refine the previous analysis. The results are reported in Table 3 with 10 large models.

In order to analyze whether our previous observations transpose to another dataset, we compute *wMPPC* on SAE features learnt on a subset of 61642 image-text pairs from Laion-2B (Schuhmann et al., 2022). At a model level, *wMPPC* values are similar to those obtained on COCO, except for comparisons between two text encoders, that obtain higher scores. However, cross modal comparisons obtain much lower *wMPPC* when considering only the last layer. As the Laion-2B captions are scraped from the internet (as opposed to the captions of COCO that are human-written for each image), this could highlight a worse image-text alignment for this dataset.

To compare models on a domain specific dataset, we compute *wMPPC* between L-size models on Oxford-102 Flowers. As this dataset has less intra-modality variability (domain specific), *wMPPC* gets higher scores than on COCO for same-modality comparisons, especially between image encoders. However, cross-modal comparisons obtain lower *wMPPC* than on COCO, suggesting a worse image-text alignment.

### 3.4 A TYPOLOGY OF VISUAL CONCEPTS SPECIFIC TO VLMS

The use of image-text contrastive learning has shown great results in understanding visual information. Then, we aim at exhibiting the gain made possible by such multimodal training, at a *concept* level, by using our SAE-based indicators. SAEs are trained on the activations resulting from the COCO dataset, holding a high image-text alignment quality (Sec 3.3). In order to identify features shared by multiple VLMs, but not by visual FMs, we compute the Generalized Comparative Sharedness $\Delta^{M \rightarrow G, H}$ (Equation 7), with CLIP features as a comparison standard $M$. For this role, we consider CLIP among all VLMs used in this study, as it is the most common, the oldest, and the least performing one. Therefore, features from CLIP that have low $\rho_i^{A \rightarrow B}$ towards other VLMs would not be caused by a performance improvement. The group $G$ comprises the visual encoders from other VLMs (SigLIP2 and DFN) as well as features from a second SAE trained on the same CLIP model as $M$, with a different seed. The group $H$ comprises the visual foundation models DinoV2, MambaVision and a ViT trained on ImageNet-21k classification.

Inspecting the features corresponding to the top 1% of $\Delta^{M \rightarrow G, H}$ (81 out of 8192), we identify the following types of concepts that are specific to VLMs (visualizations are provided in Appendix F):

- *Age related features*: among the features that are specific to all the studied VLMs, some are associated to kids in specific situations, such as birthday parties, brushing teeth or playing baseball. Each of those features is associated with a specific age range.

- *Pets having "unusual" behaviour*: the COCO dataset has lots of images of cats and dogs having unusual behaviour, such as wearing ties or hats, sitting on laptops... VLMs share multiple features associated specifically to those images, often to multiple types of those unusual behaviours, but not to classical images of pets. Visual FM don't share those features.

- *Rooms of the house* : features activated by images of a specific room of the house (bedroom, bathroom, kitchen...). In particular, some features with a high comparative sharedness are activated on images of different types of the bathroom (sink, toilet, bath). Also, those images are more cluttered than most of the COCO dataset, however relevant associations are made.

- *Vehicles*: high speed trains, fret trains and steam trains are all visually different, however they all are trains. CLIP has features activated for all those kinds of trains, and similar features for planes, cars, buses or boats are shared with all the studied VLM, but with none of the studied visual FM.

- *Old photos* : features activated for grayscale, blurry, and seemingly old photos. Even though those characteristics are purely visual, those features are specific to VLMs. Also note that recent artistic grayscale photos are present in the dataset, and have distinct features associated to them, not obtaining a high comparative sharedness.

- *Geographical features* : features activated on different kinds of images corresponding to the same geographical region. That includes features activated for multiple types of african animals (elephants, zebras and giraffes), or multiple types of Italian food (such as pastas, lasagnas and pizzas). Note that features activated only for images of zebras, or only for pizzas do not obtain a high comparative sharedness.

- *"To ride"* : one notable feature among the top 1% of $\Delta^{M \to G,H}$ is activated for images of horses, skis, snowboards, bikes, surfs or jetskis. Those are very different types of objects, but all of them are associated to the verb "to ride" in English.

Such observations confirm previous assumptions on geographical features (Stevens et al., 2025), but allows extracting a whole typology of concepts by having a more systematic approach. Most of those features seem to rely on prior knowledge, that is absent from visual foundation models without text pretraining. They are activated on images of different types of situations, corresponding to the same high-level semantic concept. In particular, the feature seemingly related to the verb "to ride" appears to rely solely on textual information, despite being extracted from a visual encoder.

### 3.5 VISUAL FEATURES SPECIFIC TO VLMS ARE REALLY *textual* FEATURES

In previous section, we present a typology of SAE features that are shared by multiple VLMs, but not by visual FMs. Then, if these specificities are a direct consequence of text pre-training, some features learnt on text encoders using image captions could have similar behaviours. We study the same CLIP image features as previously, using Generalized Comparative Sharedness $\Delta^{M \to G,H}$ to find features better shared with BERT-large and DeBERTa-large than with any of MambaVision, DinoV2 and ViT. Again, we establish a typology of concepts among the top 1% of $\Delta^{M \to G,H}$ . The features of CLIP image encoder that are better shared with every studied LLM than with any studied visual FM correspond to: *kids in a specific situation*, *rooms of the house*, *types of vehicles*, *pets having unusual behaviour* or *old photos*.

The obtained typology is very similar to the one established while considering VLM visual encoders, pushing the hypothesis that previous observations could be caused by their text pretraining. Actually, 16 features are present in the highest 81 (1%) Comparative Sharedness towards both LLMs and VLM visual encoders. Qualitative examples are represented in Figure 2, with the 9 images corresponding to the highest activations among the COCO dataset.

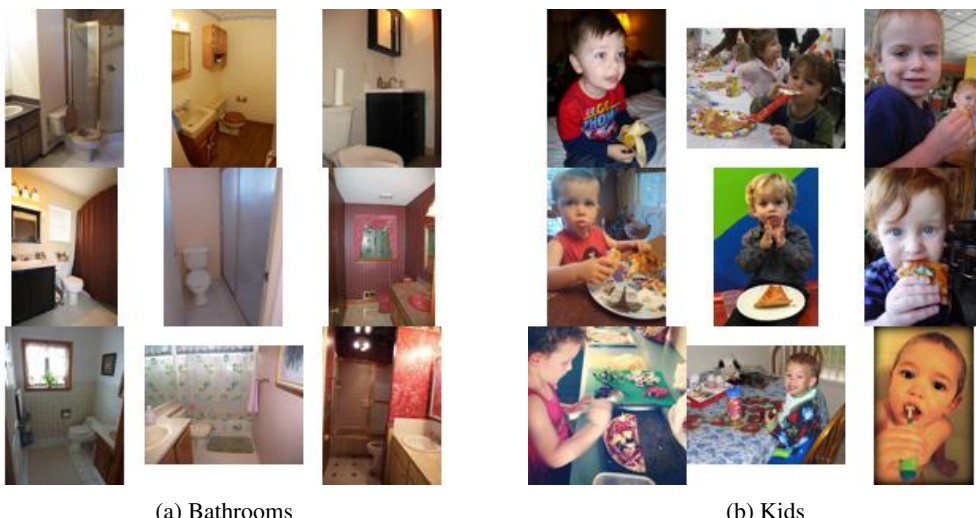

(a) Bathrooms                                                            (b) Kids

Figure 2: CLIP visual features better shared with LLMs and VLMs than with visual FMs

## 4 RELATED WORK

**Representational similarity**  As both the performance of deep neural networks improves on both text and images, recent works analyze the alignment between the representations of such networks (Kornblith et al., 2019; Klabunde et al., 2023; Boix-Adserà et al., 2022). Empirical studies (Li et al., 2024; Huh et al., 2024) find representational alignment between language and vision models, by studying the distance structure induced by their learnt vector embeddings. In particular, they find convergence of models of different architectures and modalities upon performance, suggesting the existence of a *platonic representation*.

**Universal neurons**  Analyzing individual neurons of networks has revealed neurons corresponding to interpretable features. In language models, some have been found to correspond to sentiment (Donnelly & Roegiest, 2019) or skills (Wang et al., 2022). In a similar fashion, vision models have individual neurons activated for curves with specific orientations (Cammarata et al., 2021) or specific objects (Bau et al., 2020). Multiple GPT2 models trained with different training seeds have been shown to share 1-5% of neurons (Gurnee et al., 2024), with clear interpretations and functional roles. Also, vision models trained with different tasks share *Rosetta Neurons*, activated on similar regions of images (Dravid et al., 2023). Semantic superposition is the main problem for such studies, as most neurons are polysemantic, and are activated on seemingly unrelated inputs (Elhage et al., 2022). Parekh et al. (2024) aim at learning an interpretable dictionary of multimodal concepts from a Large Multimodal Models, while Shukor & Cord (2024) study implicit multimodal alignment in frozen LLMs.

**Sparse autoencoders**  In order to disentangle the concepts corresponding to individual neurons, sparse autoencoders are trained on model activations, in order to extract sparse, and interpretable features (Bricken et al., 2023; Cunningham et al., 2023). Such features have seen promising results towards understanding language models (Lieberum et al., 2024; Gao et al., 2025; Rajamanoharan et al., 2024). Also, recent works have addressed SAEs for vision, or multimodal models (Gorton, 2024; Thasarathan et al., 2025; Lim et al., 2025; Rao et al., 2024; Papadimitriou et al., 2025; Pach et al., 2025; Yan et al., 2025; Zaigrajew et al.), in scenarios such as model adaptation (Lim et al., 2025). (FEL et al., 2023) evaluates the importance of each learnt concept, by assessing its impact on classification predictions.

**Comparing SAEs**  SAE features are used to compare different models. Universal SAEs (Thasarathan et al., 2025) learn a common concept space for three image encoders, relying on the same decoder. *MPPC* (Wang et al., 2025) performs a correlation analysis between features of two generative LLMs, in order to quantify to what extent those models share concepts. Finally, (Stevens

et al., 2025) suggests that CLIP holds visual features associated with a precise cultural or geographical context, that are absent from DinoV2.

## 5    DISCUSSION, LIMITATIONS AND PERSPECTIVES

**Discussion**    We conduct a comparative analysis of 21 visual, textual and multimodal encoders upon SAE-derived features. We introduce *wMPPC*, an indicator evaluating similarities between different models at a *concept* level, considering relative feature importance. With appropriate weights, this indicator corresponds to the expectation of the per-feature maximal correlation under sampling by activation mass and it gives a practical criterion to choose the Top-k SAE level of sparsity. From the empirical study on 21 visual and textual encoders, we find that SAEs learnt on COCO obtain higher *wMPPC* between encoders of different modalities than SAEs learnt on a subset of Laion-2B. That highlights the difference in quality of image-text alignment between the two datasets. Also, the newly defined *Comparative Sharedness* indicator allows to find individual features of a model that are better shared with a class of models than another, and to establish typologies of such features. We find that features that are specific to VLMs among vision encoders are also better shared with LLMs than visual foundation models. That emphasizes the importance of text pretraining for image understanding, by highlighting specific concepts.

**Limitations**    Although our study involves much more encoders than previous studies, all of them are based on transformers (Vaswani et al., 2017). Training SAEs on models having large and hierarchical feature maps (such as convolutional networks, or Swin transformers (Liu et al., 2021)) is possible. However, in practice such models would imply having huge SAEs, or using smaller SAEs for the largest layers (Gorton, 2024), therefore not allowing a systematic *wMPPC* analysis. One can also note that we considered only encoders while MPPC (Wang et al., 2025) focuses on language decoders. This choice resulted from the objective of studying the features shared in a multimodal context. Although visual generative models conditioned by a text could have been considered, it seemed more appropriate to first study Visual Language Models which are trained with an objective that is more symmetric between both modalities. A second limitation is the asymmetry of the proposed *wMPPC* indicator, similar to MPPC. Hence, it can not be used as an actual distance measurement between models. A naive symmetric version can be easily derived (*e.g* similarly to the Jensen–Shannon divergence with regard to the Kullback-Leibler one) but it would be at the cost of losing important information. For instance in a cross-modal context, the wMPPC is very low when SigLIP is used as source but three times larger when it is used as target (Table 1), which suggests that SigLIP encodes more concepts that are unknown by image encoders than the opposite. A symmetric version of wMPPC would not be able to highlight such a phenomenon, reporting only a bland average value of both cross-modal contexts. A final limitation identified is that even if sparse autoencoders are one of the most promising methods regarding concept-based analysis, they are not guaranteed to extract every single concept used by a model. However, regarding the tools we propose (wMPPC and Comparative Sharedness), they could be applied to future alternative methods to extract concepts as long as one could compute correlation between features of the said concepts in two models.

**Broader impact**    By addressing specifically Explainable AI (XAI) in a cross-modal context, this paper can contribute to transfer representation from one modality to another. It can also contribute to improve a user's understanding of the inner structure of a large model, by providing explanations through multiple modalities.

**Perspectives**    Our findings highlight that *wMPPC* can be used to assess the quality of the image-text alignment of a dataset. Comparative studies of multiple image-text datasets could be performed, in order to select or filter datasets used for training new models. Techniques for automatically naming SAE features considering both images and captions could allow large scale *Comparative Sharedness* analysis, using features of both modalities as comparison standards. All the models considered in this work are encoder models. As SAE-derived features have been studied extensively for models specialized in text generation, a systematic analysis of *wMPPC* on generative models with different modalities could provide meaningful insight into their behaviour.

## REPRODUCIBILITY STATEMENT

Our results can be reproduced, following the method described in section 2 and subsection 3.1. Corresponding code is provided as supplemental material. It is based on free software and libraries (see Appendix G for the licences). Links and details to the models are provided in Appendix A (Table 4 and Table 5).

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

## A APPENDIX: ENCODER DESCRIPTION

We report in Table 4 all the encoders considered in our study with their key features and the link to download them in Table 5. All the models were downloaded from huggingface, except from CLIP and DFN models from OpenClip (Ilharco et al., 2021b) and DinoV2 from PyTorch Hub. The *model size* is the number of parameters and since all of them were encoded in `float32` their actual size in memory is this number multiplied by four. Six datasets were used to train them.

CLIP was trained "on publicly available image-caption data" that is image-caption pairs from the Web and publicly available datasets such as YFCC 100M (Thomee et al., 2016). The creator of the model did not release the dataset to avoid its use "as the basis for any commercial or deployed model".

DFN is a CLIP-like model trained from 2B image-text pairs, resulting from the filtering of a pool of 12.8 billion uncurated image-text pairs of CommonPool, collected from Common Crawl. This last is itself part of DataComp, a benchmark for designing multimodal datasets (Gadre et al., 2023).

MambaVision and ViT were trained on well-known and publicly available ImageNet dataset (Deng et al., 2009) with $1,000$ categories of the Large Scale Visual Recognition Challenge (Russakovsky et al., 2015) or the full $21k$ classes.

In Table 3:

- the set "L" contains 10 encoders: CLIP ViT L/14 (both image and text encoders), DFN ViT L/14 (both image and text encoders), SigLIP2 L/16 (both image and text encoders), DinoV2 L/14 (image encoder), ViT L/16 (image encoder), BERT large (text encoder) and DeBERTa (text encoder)

- the set "B" contains 10 encoders: CLIP ViT B/32 (both image and text encoders), DFN ViT B/16 (both image and text encoder), SigLIP2 B/16 (both image and text encoder), DinoV2 B/14 (image encoder), ViT B/16 (image encoder), BERT base (text encoder) and DeBERTa base (text encoder)

Table 4: Pre-trained encoders considered in this study.

| Encoder | input type | Model type | Model size | Training set |
|---|---|---|---|---|
| CLIP ViT B/32 (Radford et al., 2021) | image+text | VLM | 87M | openAI private: web, YFCC100M... |
| CLIP ViT L/14 (Radford et al., 2021) | image+text | VLM | 303M | |
| DFN ViT B/16 (Fang et al., 2024) | image+text | VLM | 86M | 2B filtered from CommonPool -12.8B |
| DFN ViT L/14 (Fang et al., 2024) | image+text | VLM | 303M | |
| SigLIP2 B/16 (Tschannen et al., 2025) | image+text | VLM | 92M | WebLI |
| SigLIP2 L/16 (Tschannen et al., 2025) | image+text | VLM | 316M | |
| DinoV2 B/14 (Oquab et al., 2023) | image | visual FM | 86M | LVD-142M |
| DinoV2 L/14 (Oquab et al., 2023) | image | visual FM | 304M | |
| MambaVision B (Hatamizadeh & Kautz, 2024) | image | visual FM | 97M | ImageNet-1k ImageNet-21k |
| ViT B/16 (Dosovitskiy et al., 2020) | image | visual FM | 86M | |
| ViT L/16 (Dosovitskiy et al., 2020) | image | visual FM | 303M | |
| BERT base (Devlin et al., 2019) | text | LLM | 109M | BookCorpus, Wikipedia |
| BERT large (Devlin et al., 2019) | text | LLM | 335M | |
| DeBERTa base (He et al., 2021) | text | LLM | 99M | BookCorpus, Wikipedia, OpenWebText, STORIES |
| DeBERTa large (He et al., 2021) | text | LLM | 353M | |

## B THEORETICAL DERIVATION

In order to compare different models upon their SAE features we generalize the MPPC indicator (Wang et al., 2025).

The MPPC per-feature of the $i$-th SAE feature $f_i^A$ learnt for a pretrained model $A$ is defined by its maximum pairwise Pearson correlation among features of model $B$ and is noted $\rho_i^{A \to B}$:

$$\rho_i^{A \to B} = \max_j \frac{\mathbb{E}[(\boldsymbol{f}_i^A - \mu_i^A)(\boldsymbol{f}_j^B - \mu_j^B)]}{\sigma_i^A \sigma_j^B} \tag{8}$$

with $\boldsymbol{f}_i^A, \boldsymbol{f}_j^B$ the $i$-th feature of $A$ and the $j$-th feature of $B$, $\mu_i^A, \mu_j^B$ their respective means, $\sigma_i^A, \sigma_j^B$ their standard deviations. In practice the correlations are estimated with $N$ sample data.

With nonnegative weights $w_i$ such that $\sum_{i=1}^n w_i = 1$, the weighted MPPC is:

$$\text{wMPPC} = \sum_{i=1}^n w_i \rho_i^{A \to B} \tag{2}$$

Let us consider $a_i(x) \in \mathbb{R}$ the activation (nonnegative scalar because of the ReLU) of the SAE feature $f_i^A$ on input $x \in \mathcal{D}$, and the cumulative activation of feature $i$ over $\mathcal{D}$ noted as $S_i^A = \sum_{x \in \mathcal{D}} a_i(x)$.

**Proposition 1.** *If one considers the normalized weights $w_i = \frac{S_i^A}{\sum_{\ell=1}^n S_\ell^A}$ we have:*

$$\text{wMPPC}^{A \to B} = \frac{1}{\sum_{\ell=1}^n S_\ell^A} \sum_{x \in \mathcal{D}} \sum_{i=1}^n a_i(x) \rho_i^{A \to B} \tag{3}$$

*Therefore $\text{wMPPC}^{A \to B} = \mathbb{E}_g\left[\rho_i^{A \to B}\right]$ where $g$ is the joint distribution that samples a datapoint $x$ in $\mathcal{D}$ and then samples feature $i$ with probability proportional to $a_i(x)$.*

*Proof.* Let substitute $w_i = S_i^A / \sum_\ell S_\ell^A$ into Equation 2:

$$\text{wMPPC} = \sum_{i=1}^n \frac{S_i^A}{\sum_\ell S_\ell^A} \rho_i^{A \to B} \tag{9}$$

$$= \frac{1}{\sum_\ell S_\ell^A} \sum_{i=1}^n \left( \sum_{x \in \mathcal{D}} a_i(x) \right) \rho_i^{A \to B} \tag{10}$$

$$= \frac{1}{\sum_\ell S_\ell^A} \sum_{x \in \mathcal{D}} \sum_{i=1}^n a_i(x) \rho_i^{A \to B} \tag{11}$$

Thus if one samples $x$ with uniform probability over $\mathcal{D}$ then, conditional on $x$, let pick feature $i$ with probability proportional to $a_i(x)$. The inner sum divided by $\sum_\ell c_\ell$ is exactly the expected value of $\rho_i^{A \to B}$ under that two-stage sampling. $\qquad \square$

If we use TopK-SAE, let us consider the order statistics of the activations $a_i^{(1)} \geq a_i^{(2)} \geq \cdots \geq a_i^{(k)}$ then define the Top-$k$ cumulative activation as

$$c_i^{(k)} = \sum_{t=1}^k a_i^{(t)} \tag{12}$$

Let denote the resulting sum of cumulative activations as:

$$C^{(k)} = \sum_{i=1}^n c_i^{(k)} \tag{13}$$

The Top-$k$ activation-weighted MPPC is then defined as:

$$\text{wMPPC}_{(k)}^{A \to B} = \frac{1}{C^{(k)}} \sum_{i=1}^n c_i^{(k)} \rho_i^{A \to B} \tag{14}$$

**Proposition 2.** *(incremental update) Under the above notation and assumption,*

$$\text{wMPPC}^{A\to B}_{(k+1)} - \text{wMPPC}^{A\to B}_{(k)} = \frac{1}{C^{(k+1)}} \sum_{i=1}^{n} a_i^{(k+1)} \big(\rho_i^{A\to B} - \text{MPPC}^{A\to B}_{(k)}\big) \qquad (15)$$

*Proof.* We want to determine the effect of adding a $k+1$-th activation $a_i^{(k+1)}$ in the Top-$k$ SAE on $\text{wMPPC}^{A\to B}_{(k)}$. Let us note $A^{(k+1)} = \sum_{i=1}^{n} a_i^{(k+1)}$, we have:

$$C^{(k+1)} = \sum_{i=1}^{n}\sum_{t=1}^{k+1} a_i^{(t)} = \sum_{i=1}^{n}\sum_{t=1}^{k} a_i^{(t)} + \sum_{i=1}^{n} a_i^{(k+1)} = C^{(k)} + A^{(k+1)} \qquad (16)$$

Using the definition of $\text{wMPPC}^{A\to B}_{(k)}$ (Equation 14) we have:

$$\text{wMPPC}^{A\to B}_{(k+1)} - \text{wMPPC}^{A\to B}_{(k)} = \frac{\sum_i (c_i^{(k)} + a_i)\rho_i^{A\to B}}{C^{(k)} + S} - \frac{\sum_i c_i^{(k)}\rho_i^{A\to B}}{C^{(k)}} \qquad (17)$$

$$= \underbrace{\frac{\sum_i c_i^{(k)}\rho_i^{A\to B}}{C^{(k)} + S} - \frac{\sum_i c_i^{(k)}\rho_i^{A\to B}}{C^{(k)}}}_{=:T_1} + \underbrace{\frac{\sum_i a_i\rho_i^{A\to B}}{C^{(k)} + A^{(k+1)}}}_{=:T_2} \qquad (18)$$

The first term $T_1$ can be reformulated as:

$$T_1 = \sum_i c_i^{(k)}\rho_i^{A\to B} \cdot \Big(\frac{1}{C^{(k)} + A^{(k+1)}} - \frac{1}{C^{(k)}}\Big) \qquad (19)$$

$$= \sum_i c_i^{(k)}\rho_i^{A\to B} \cdot \Big(\frac{C^{(k)} - (C^{(k)} + A^{(k+1)})}{C^{(k)}(C^{(k)} + A^{(k+1)})}\Big) \qquad (20)$$

$$= -\frac{A^{(k+1)}}{C^{(k)}(C^{(k)} + A^{(k+1)})} \sum_i c_i^{(k)}\rho_i^{A\to B} \qquad (21)$$

Thus:

$$\text{wMPPC}^{A\to B}_{(k+1)} - \text{wMPPC}^{A\to B}_{(k)} = \frac{\sum_i a_i\rho_i^{A\to B}}{C^{(k)} + A^{(k+1)}} - \frac{A^{(k+1)}}{C^{(k)}(C^{(k)} + A^{(k+1)})} \sum_i c_i^{(k)}\rho_i^{A\to B} \qquad (22)$$

$$= \frac{1}{C^{(k)} + A^{(k+1)}}\Big(\sum_i a_i\rho_i^{A\to B} - A^{(k+1)}\,\text{wMPPC}^{A\to B}_{(k)}\Big) \qquad (23)$$

$$= \frac{1}{C^{(k+1)}} \sum_{i=1}^{n} a_i\big(\rho_i^{A\to B} - \text{wMPPC}^{A\to B}_{(k)}\big) \qquad (24)$$

by using $\sum_i c_i^{(k)}\rho_i^{A\to B} = C^{(k)}\,\text{wMPPC}^{A\to B}_{(k)}$ (according to Equation 14) to go from Equation 22 to Equation 23. $\qquad\square$

From Proposition 2 we thus have the results of the main paper:

**Corollary 1.** *(monotonicity condition) A sufficient condition for* $\text{wMPPC}^{A\to B}_{(k+1)} \geq \text{wMPPC}^{A\to B}_{(k)}$ *is:*

$$\sum_{i=1}^{n} a_i^{(k+1)}\big(\rho_i^{A\to B} - \text{MPPC}^{(k)}_w\big) \geq 0 \qquad (5)$$

## C  APPENDIX: STATISTICAL SIGNIFICANCE OF MPPC AND wMPPC

With $\rho_i$ the maximum pairwise coefficient (Equation 8) for $N$ target features of length $L$, and $H_0$ the hypothesis of features having no linear relationship. Using the Fischer z-transformation (Fisher, 1915)

$$z = artanh(r) \sim \mathcal{N}(0, \frac{1}{\sqrt{L-3}})$$

$$\mathbb{P}(\max_i(r_i) > x) = 1 - \mathbb{P}(r \leq x)^N$$

$$\mathbb{P}(\rho_i > x) = \mathbb{P}(\max_i(z_i) > artanh(x))$$

$$\mathbb{P}(\rho_i > x) = 1 - \Phi(artanh(x)\sqrt{L-3})^N$$

With $N = 8192$ (corresponding to ViT-L experiments, for one layer), and $L = 10000$ being largely lower than the size of the COCO dataset used, we obtain $\mathbb{P}(\rho_i > 0.3) \approx 10^{-206}$ , thus reject $H_0$.

Experimentally, for two sparse autoencoders trained on the same CLIP-ViT-B/32 visual encoder, on the COCO dataset and shuffling the features upon images (to preserve the density of feature distributions), we obtain *wMPPC* $= 0.0125$, while the non-shuffled *wMPPC* $= 0.5854$.

## D   APPENDIX: DETAILED WMPPC RESULTS

In Table 3 we report average results of wMPPC that provide an overview to the reader for various settings. In this appendix, we report the detailed results of each setting that led to these average scores. The setting are "multidimensional" thus we provide in Table 6 the synthetic pointers to help the reading.

Also, note that MambaVision is only considered at its last layer on COCO (Table 9 and Table 10), as it is only used to compute Comparative Sharedness.

## E   APPENDIX: MPPC RESULTS

We report results using MPPC (Wang et al., 2025) to compare large versions of encoders on the Flowers-102 dataset in Table 23, Table 24, Table 25, and Table 26. Observed similarities tend to be closer to each other than those obtained with *wMPPC*, as no weighting towards features of interrest is used.

## F   APPENDIX: ADDITIONAL EXAMPLES OF VISUAL FEATURES SPECIFIC TO VLMS

In subsection 3.4, we provide a typology of features learnt on CLIP visual encoder that are better share with other VLMs than with visual FMs. Figure 3 contains an example for each mentioned category. We display the feature corresponding to the highest Generalized Comparative Sharedness for each category, except for features present in Figure 2. In Figure 4, we represent the 100 images corresponding to the highest activations of the feature associated to the verb "to ride". A larger number of examples is chosen here, in order to better represent the diversity of objects that activate this particular feature.

## G   APPENDIX: CODE ASSOCIATED TO THE PAPER

The code to reproduce the experiments is provided as supplementary material (zip file). It is developed from scratch and relies mainly on PyTorch (Paszke et al., 2019) and numpy (Harris et al., 2020).

We used OpenCLIP (Ilharco et al., 2021a) and the Huggingface Transformers library (Wolf et al., 2020) to handle models. As well, we relied on the Huggingface Datasets library (Lhoest et al., 2021) to handle the datasets.

All these libraries are open source with permissive software license, as summarized in Table 27

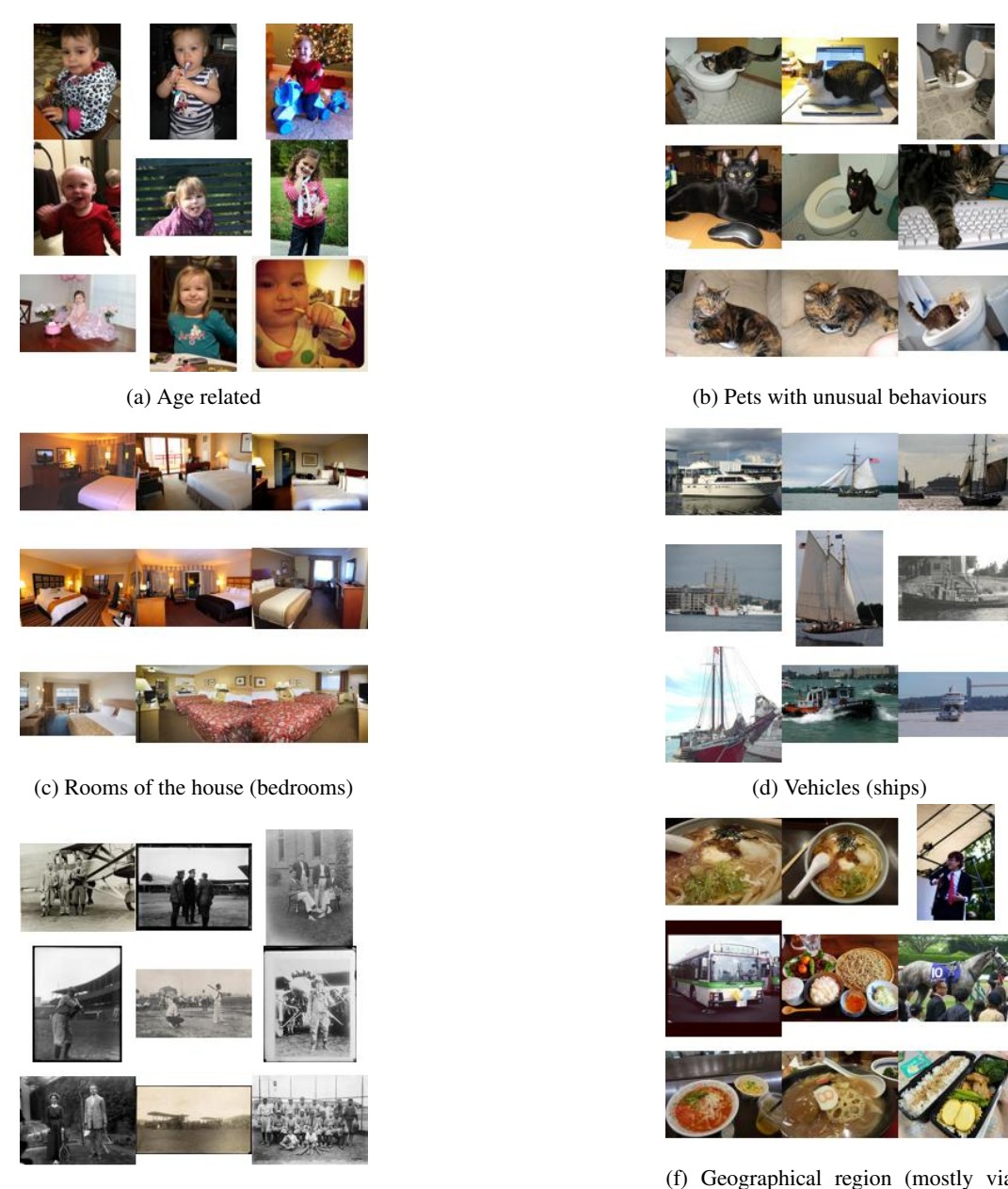

(a) Age related

(b) Pets with unusual behaviours

(c) Rooms of the house (bedrooms)

(d) Vehicles (ships)

(e) Old photos

(f) Geographical region (mostly via food)

Figure 3: Examples of visual features specific to VLMs mentioned in subsection 3.4

## H APPENDIX: LLM USAGE

Beyond the usage of LLM described in the paper, that is part of the study, we used commercial services to polish the writting: find synonyms, rephrase sentences.

We also used such an LLM service to conduct an initial investigation into certain theoretical derivations (Appendix B). The final proofs were established and asserted by ourselves.

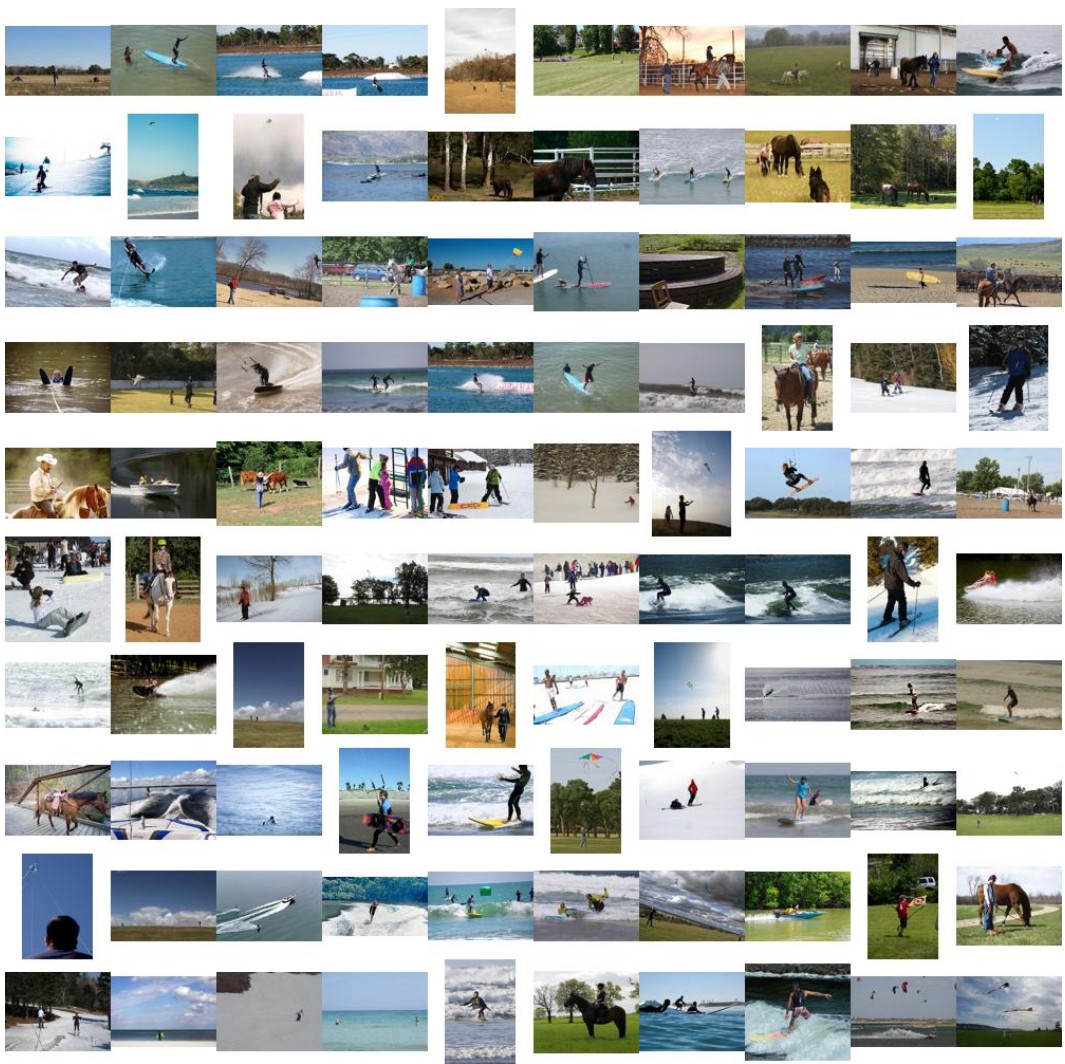

Figure 4: 100 images corresponding to the highest activations of the feature associated to the verb "to ride"

Table 5: Links to the pre-trained encoders considered in this study.

| Encoder | URL |
|---------|-----|
| CLIP ViT B/32  (Radford et al., 2021) | ⬇ |
| CLIP ViT L/14  (Radford et al., 2021) | ⬇ |
| DFN ViT B/16  (Fang et al., 2024) | ⬇ |
| DFN ViT L/14  (Fang et al., 2024) | ⬇ |
| SigLIP2 B/16  (Tschannen et al., 2025) | ⬇ |
| SigLIP2 L/16  (Tschannen et al., 2025) | ⬇ |
| DinoV2 B/14  (Oquab et al., 2023) | ⬇ |
| DinoV2 L/14  (Oquab et al., 2023) | ⬇ |
| MambaVision B  (Hatamizadeh & Kautz, 2024) | ⬇ |
| ViT B/16 (Dosovitskiy et al., 2020) | ⬇ |
| ViT L/16  (Dosovitskiy et al., 2020) | ⬇ |
| BERT base  (Devlin et al., 2019) | ⬇ |
| BERT large  (Devlin et al., 2019) | ⬇ |
| DeBERTa base  (He et al., 2021) | ⬇ |
| DeBERTa large  (He et al., 2021) | ⬇ |

Table 6: Synthetic pointers to the tables of detailed results. It gives the table according to the input dataset and the size of the encoders (each row), the modality of the target encoders that can be 'Image' (col 3) or 'Text' (col 4) and finally whether 'all' layers or only the 'last' one is used to compute wMPPC.

| Dataset | Model size | Target encoders | |
|---------|-----------|-----------------|---|
| | | Image | Text |
| COCO | large | Table 7 (all) Table 9 (last) | Table 8 (all) Table 10 (last) |
| | base | Table 11 (all) Table 13 (last) | Table 12 (all) Table 14 (last) |
| LAION | large | Table 15 (all) Table 17 (last) | Table 16 (all) Table 18 (last) |
| Flowers-102 | large | Table 19 (all) Table 21 (last) | Table 20 (all) Table 22 (last) |

Table 7: $wMPPC^{source \rightarrow target}$ (all layers) on COCO, for all 10 large models as source, image encoders as target

| Target / Source | | Image | | | | |
|---|---|---|---|---|---|---|
| | | CLIP (I) | SigLIP2 (I) | DFN (I) | DinoV2 | ViT |
| Image | CLIP (I) | 1 | 0.446 | 0.489 | 0.486 | 0.444 |
| | SigLIP2 (I) | 0.514 | 1 | 0.516 | 0.509 | 0.500 |
| | DFN (I) | 0.469 | 0.416 | 1 | 0.431 | 0.385 |
| | DinoV2 | 0.556 | 0.518 | 0.533 | 1 | 0.515 |
| | ViT | 0.390 | 0.381 | 0.379 | 0.391 | 1 |
| Text | CLIP (T) | 0.253 | 0.275 | 0.254 | 0.246 | 0.223 |
| | SigLIP2 (T) | 0.045 | 0.051 | 0.044 | 0.043 | 0.037 |
| | DFN (T) | 0.248 | 0.282 | 0.257 | 0.235 | 0.227 |
| | BERT | 0.182 | 0.195 | 0.181 | 0.177 | 0.158 |
| | DeBERTa | 0.119 | 0.129 | 0.120 | 0.113 | 0.105 |

Table 8: *wMPPC$^{source \rightarrow target}$* (all layers) on COCO, for all 10 large models as source, text encoders as target

| | Target
Source | Text
CLIP (T) | SigLIP2 (T) | DFN (T) | BERT | DeBERTa |
|---|---|---|---|---|---|---|
| **Image** | CLIP (I) | 0.209 | 0.131 | 0.214 | 0.194 | 0.188 |
| | SigLIP2 (I) | 0.272 | 0.171 | 0.282 | 0.251 | 0.248 |
| | DFN (I) | 0.203 | 0.128 | 0.208 | 0.188 | 0.182 |
| | DinoV2 | 0.250 | 0.153 | 0.256 | 0.233 | 0.224 |
| | ViT | 0.201 | 0.132 | 0.207 | 0.186 | 0.183 |
| **Text** | CLIP (T) | 1 | 0.351 | 0.509 | 0.428 | 0.412 |
| | SigLIP2 (T) | 0.256 | 1 | 0.254 | 0.578 | 0.400 |
| | DFN (T) | 0.480 | 0.327 | 1 | 0.426 | 0.431 |
| | BERT | 0.346 | 0.287 | 0.352 | 1 | 0.361 |
| | DeBERTa | 0.266 | 0.256 | 0.274 | 0.344 | 1 |

Table 9: *wMPPC$^{source \rightarrow target}$* (last layer) on COCO, for all 10 large models and MambaVision as source, image encoders as target

| | Target
Source | Image
CLIP (I) | SigLIP2 (I) | DFN (I) | DinoV2 | ViT | MambaVision |
|---|---|---|---|---|---|---|---|
| **Image** | CLIP (I) | 1 | 0.278 | 0.265 | 0.208 | 0.232 | 0.215 |
| | SigLIP2 (I) | 0.320 | 1 | 0.329 | 0.236 | 0.293 | 0.294 |
| | DFN (I) | 0.267 | 0.287 | 1 | 0.207 | 0.236 | 0.225 |
| | DinoV2 | 0.270 | 0.290 | 0.277 | 1 | 0.270 | 0.259 |
| | ViT | 0.258 | 0.286 | 0.272 | 0.226 | 1 | 0.264 |
| | MambaVision | 0.236 | 0.281 | 0.258 | 0.214 | 0.264 | 1 |
| **Text** | CLIP (T) | 0.255 | 0.284 | 0.265 | 0.211 | 0.252 | 0.234 |
| | SigLIP2 (T) | 0.054 | 0.062 | 0.054 | 0.042 | 0.049 | 0.048 |
| | DFN (T) | 0.246 | 0.287 | 0.258 | 0.196 | 0.245 | 0.227 |
| | BERT | 0.183 | 0.195 | 0.179 | 0.136 | 0.168 | 0.154 |
| | DeBERTa | 0.150 | 0.162 | 0.149 | 0.115 | 0.139 | 0.128 |

Table 10: *wMPPC$^{source \rightarrow target}$* (last layer) on COCO, for all 10 large models as source, text encoders as target

| | Target
Source | Text
CLIP (T) | SigLIP2 (T) | DFN (T) | BERT | DeBERTa |
|---|---|---|---|---|---|---|
| **Image** | CLIP (I) | 0.220 | 0.128 | 0.218 | 0.203 | 0.207 |
| | SigLIP2 (I) | 0.274 | 0.153 | 0.278 | 0.249 | 0.256 |
| | DFN (I) | 0.222 | 0.127 | 0.223 | 0.199 | 0.203 |
| | DinoV2 | 0.254 | 0.142 | 0.260 | 0.216 | 0.226 |
| | ViT | 0.243 | 0.130 | 0.248 | 0.215 | 0.223 |
| | MambaVision | 0.219 | 0.117 | 0.223 | 0.190 | 0.197 |
| **Text** | CLIP (T) | 1 | 0.192 | 0.361 | 0.286 | 0.306 |
| | SigLIP2 (T) | 0.134 | 1 | 0.102 | 0.297 | 0.347 |
| | DFN (T) | 0.345 | 0.173 | 1 | 0.282 | 0.301 |
| | BERT | 0.237 | 0.172 | 0.238 | 1 | 0.275 |
| | DeBERTa | 0.229 | 0.195 | 0.226 | 0.288 | 1 |

Table 11: *wMPPC*$^{source \rightarrow target}$ (all layers) on COCO, for all 10 base models as source, image encoders as target

| Target | | CLIP (I) | SigLIP2 (I) | DFN (I) | DinoV2 | ViT |
|--------|--------|----------|-------------|---------|--------|------|
| Source | | Image | | | | |
| Image | CLIP (I) | 1 | 0.487 | 0.530 | 0.504 | 0.485 |
| | SigLIP2 (I) | 0.581 | 1 | 0.584 | 0.562 | 0.579 |
| | DFN (I) | 0.527 | 0.500 | 1 | 0.522 | 0.485 |
| | DinoV2 | 0.499 | 0.499 | 0.523 | 1 | 0.481 |
| | ViT | 0.459 | 0.458 | 0.465 | 0.455 | 1 |
| Text | CLIP (T) | 0.227 | 0.261 | 0.250 | 0.252 | 0.229 |
| | SigLIP2 (T) | 0.071 | 0.077 | 0.073 | 0.073 | 0.068 |
| | DFN (T) | 0.240 | 0.281 | 0.269 | 0.270 | 0.251 |
| | BERT | 0.154 | 0.171 | 0.167 | 0.168 | 0.152 |
| | DeBERTa | 0.145 | 0.159 | 0.155 | 0.157 | 0.140 |

Table 12: *wMPPC*$^{source \rightarrow target}$ (all layers) on COCO, for all 10 base models as source, text encoders as target

| Target | | CLIP (T) | SigLIP2 (T) | DFN (T) | BERT | DeBERTa |
|--------|--------|----------|-------------|---------|------|---------|
| Source | | Text | | | | |
| Image | CLIP (I) | 0.240 | 0.150 | 0.241 | 0.213 | 0.207 |
| | SigLIP2 (I) | 0.273 | 0.174 | 0.275 | 0.241 | 0.237 |
| | DFN (I) | 0.255 | 0.158 | 0.261 | 0.227 | 0.220 |
| | DinoV2 | 0.277 | 0.177 | 0.282 | 0.245 | 0.240 |
| | ViT | 0.232 | 0.148 | 0.237 | 0.207 | 0.202 |
| Text | CLIP (T) | 1 | 0.343 | 0.508 | 0.408 | 0.399 |
| | SigLIP2 (T) | 0.363 | 1 | 0.361 | 0.479 | 0.433 |
| | DFN (T) | 0.577 | 0.408 | 1 | 0.466 | 0.457 |
| | BERT | 0.358 | 0.320 | 0.359 | 1 | 0.442 |
| | DeBERTa | 0.323 | 0.330 | 0.324 | 0.437 | 1 |

Table 13: *wMPPC*$^{source \rightarrow target}$ (last layer) on COCO, for all 10 base models as source, image encoders as target

| Target | | CLIP (I) | SigLIP2 (I) | DFN (I) | DinoV2 | ViT |
|--------|--------|----------|-------------|---------|--------|------|
| Source | | Image | | | | |
| Image | CLIP (I) | 1 | 0.353 | 0.335 | 0.266 | 0.282 |
| | SigLIP2 (I) | 0.352 | 1 | 0.356 | 0.278 | 0.304 |
| | DFN (I) | 0.293 | 0.318 | 1 | 0.239 | 0.255 |
| | DinoV2 | 0.217 | 0.257 | 0.251 | 1 | 0.250 |
| | ViT | 0.245 | 0.275 | 0.263 | 0.228 | 1 |
| Text | CLIP (T) | 0.224 | 0.260 | 0.247 | 0.208 | 0.227 |
| | SigLIP2 (T) | 0.084 | 0.095 | 0.089 | 0.075 | 0.080 |
| | DFN (T) | 0.232 | 0.276 | 0.269 | 0.227 | 0.242 |
| | BERT | 0.180 | 0.192 | 0.184 | 0.139 | 0.158 |
| | DeBERTa | 0.156 | 0.164 | 0.152 | 0.112 | 0.130 |

Table 14: $wMPPC^{source \to target}$ (last layer) on COCO, for all 10 base models as source, text encoders as target

| | Target / Source | Text | | | | |
|---|---|---|---|---|---|---|
| | | CLIP (T) | SigLIP2 (T) | DFN (T) | BERT | DeBERTa |
| Image | CLIP (I) | 0.275 | 0.163 | 0.275 | 0.268 | 0.251 |
| | SigLIP2 (I) | 0.296 | 0.166 | 0.293 | 0.281 | 0.264 |
| | DFN (I) | 0.240 | 0.138 | 0.242 | 0.223 | 0.210 |
| | DinoV2 | 0.235 | 0.132 | 0.238 | 0.195 | 0.178 |
| | ViT | 0.232 | 0.127 | 0.230 | 0.209 | 0.195 |
| Text | CLIP (T) | 1 | 0.190 | 0.345 | 0.284 | 0.268 |
| | SigLIP2 (T) | 0.205 | 1 | 0.214 | 0.273 | 0.282 |
| | DFN (T) | 0.419 | 0.237 | 1 | 0.342 | 0.324 |
| | BERT | 0.244 | 0.193 | 0.271 | 1 | 0.338 |
| | DeBERTa | 0.208 | 0.261 | 0.218 | 0.388 | 1 |

Table 15: $wMPPC^{source \to target}$ (all layers) on Laion, for all 10 large models as source, image encoders as target

| | Target / Source | Image | | | | |
|---|---|---|---|---|---|---|
| | | CLIP (I) | SigLIP2 (I) | DFN (I) | DinoV2 | ViT |
| Image | CLIP (I) | 1 | 0.471 | 0.507 | 0.506 | 0.464 |
| | SigLIP2 (I) | 0.531 | 1 | 0.519 | 0.526 | 0.515 |
| | DFN (I) | 0.428 | 0.379 | 1 | 0.409 | 0.365 |
| | DinoV2 | 0.566 | 0.531 | 0.551 | 1 | 0.532 |
| | ViT | 0.401 | 0.394 | 0.387 | 0.409 | 1 |
| Text | CLIP (T) | 0.174 | 0.190 | 0.177 | 0.159 | 0.138 |
| | SigLIP2 (T) | 0.089 | 0.099 | 0.091 | 0.087 | 0.073 |
| | DFN (T) | 0.194 | 0.212 | 0.196 | 0.174 | 0.152 |
| | BERT | 0.148 | 0.152 | 0.140 | 0.133 | 0.113 |
| | DeBERTa | 0.127 | 0.134 | 0.126 | 0.114 | 0.096 |

Table 16: $wMPPC^{source \to target}$ (all layers) on Laion, for all 10 large models as source, text encoders as target

| | Target / Source | Text | | | | |
|---|---|---|---|---|---|---|
| | | CLIP (T) | SigLIP2 (T) | DFN (T) | BERT | DeBERTa |
| Image | CLIP (I) | 0.162 | 0.072 | 0.186 | 0.151 | 0.136 |
| | SigLIP2 (I) | 0.215 | 0.093 | 0.253 | 0.201 | 0.181 |
| | DFN (I) | 0.137 | 0.065 | 0.158 | 0.128 | 0.116 |
| | DinoV2 | 0.171 | 0.075 | 0.198 | 0.161 | 0.143 |
| | ViT | 0.145 | 0.063 | 0.171 | 0.139 | 0.122 |
| Text | CLIP (T) | 1 | 0.582 | 0.663 | 0.583 | 0.547 |
| | SigLIP2 (T) | 0.621 | 1 | 0.613 | 0.696 | 0.732 |
| | DFN (T) | 0.590 | 0.482 | 1 | 0.504 | 0.480 |
| | BERT | 0.389 | 0.342 | 0.381 | 1 | 0.392 |
| | DeBERTa | 0.446 | 0.482 | 0.435 | 0.520 | 1 |

Table 17: $wMPPC^{source \rightarrow target}$ (last layer) on Laion, for all 10 large models as source, image encoders as target

| Target Source | | CLIP (I) | SigLIP2 (I) | DFN (I) | DinoV2 | ViT |
|---|---|---|---|---|---|---|
| Image | CLIP (I) | 1 | 0.228 | 0.215 | 0.129 | 0.172 |
| | SigLIP2 (I) | 0.252 | 1 | 0.244 | 0.150 | 0.212 |
| | DFN (I) | 0.204 | 0.215 | 1 | 0.131 | 0.214 |
| | DinoV2 | 0.130 | 0.150 | 0.131 | 1 | 0.137 |
| | ViT | 0.212 | 0.242 | 0.214 | 0.158 | 1 |
| Text | CLIP (T) | 0.148 | 0.153 | 0.139 | 0.084 | 0.114 |
| | SigLIP2 (T) | 0.088 | 0.092 | 0.083 | 0.042 | 0.058 |
| | DFN (T) | 0.180 | 0.192 | 0.180 | 0.101 | 0.139 |
| | BERT | 0.171 | 0.170 | 0.160 | 0.091 | 0.130 |
| | DeBERTa | 0.130 | 0.138 | 0.123 | 0.068 | 0.090 |

Table 18: $wMPPC^{source \rightarrow target}$ (last layer) on Laion, for all 10 large models as source, text encoders as target

| Target Source | | CLIP (T) | SigLIP2 (T) | DFN (T) | BERT | DeBERTa |
|---|---|---|---|---|---|---|
| Image | CLIP (I) | 0.139 | 0.067 | 0.155 | 0.158 | 0.140 |
| | SigLIP2 (I) | 0.156 | 0.073 | 0.182 | 0.181 | 0.154 |
| | DFN (I) | 0.123 | 0.061 | 0.145 | 0.135 | 0.117 |
| | DinoV2 | 0.086 | 0.040 | 0.098 | 0.093 | 0.083 |
| | ViT | 0.133 | 0.058 | 0.141 | 0.145 | 0.124 |
| Text | CLIP (T) | 1 | 0.146 | 0.231 | 0.198 | 0.190 |
| | SigLIP2 (T) | 0.483 | 1 | 0.379 | 0.553 | 0.674 |
| | DFN (T) | 0.238 | 0.147 | 1 | 0.217 | 0.215 |
| | BERT | 0.221 | 0.166 | 0.221 | 1 | 0.256 |
| | DeBERTa | 0.412 | 0.392 | 0.398 | 0.431 | 1 |

Table 19: $wMPPC^{source \rightarrow target}$ (all layers) on Flowers-102, for all 10 large models as source, image encoders as target

| Target Source | | CLIP (I) | SigLIP2 (I) | DFN (I) | DinoV2 | ViT |
|---|---|---|---|---|---|---|
| Image | CLIP (I) | 1 | 0.534 | 0.567 | 0.557 | 0.518 |
| | SigLIP2 (I) | 0.666 | 1 | 0.670 | 0.659 | 0.657 |
| | DFN (I) | 0.519 | 0.479 | 1 | 0.493 | 0.452 |
| | DinoV2 | 0.598 | 0.575 | 0.591 | 1 | 0.575 |
| | ViT | 0.457 | 0.466 | 0.456 | 0.480 | 1 |
| Text | CLIP (T) | 0.132 | 0.143 | 0.133 | 0.138 | 0.135 |
| | SigLIP2 (T) | 0.083 | 0.093 | 0.089 | 0.101 | 0.088 |
| | DFN (T) | 0.161 | 0.170 | 0.162 | 0.167 | 0.164 |
| | BERT | 0.121 | 0.147 | 0.134 | 0.146 | 0.133 |
| | DeBERTa | 0.108 | 0.122 | 0.116 | 0.125 | 0.121 |

Table 20: $wMPPC^{source \rightarrow target}$ (all layers) on Flowers-102, for all 10 large models as source, text encoders as target

| | Target / Source | Text | | | | |
|---|---|---|---|---|---|---|
| | | CLIP (T) | SigLIP2 (T) | DFN (T) | BERT | DeBERTa |
| Image | CLIP (I) | 0.240 | 0.095 | 0.259 | 0.154 | 0.134 |
| | SigLIP2 (I) | 0.238 | 0.100 | 0.256 | 0.157 | 0.141 |
| | DFN (I) | 0.200 | 0.086 | 0.215 | 0.133 | 0.117 |
| | DinoV2 | 0.291 | 0.107 | 0.313 | 0.184 | 0.159 |
| | ViT | 0.247 | 0.097 | 0.266 | 0.160 | 0.142 |
| Text | CLIP (T) | 1 | 0.440 | 0.630 | 0.516 | 0.488 |
| | SigLIP2 (T) | 0.508 | 1 | 0.503 | 0.462 | 0.465 |
| | DFN (T) | 0.620 | 0.419 | 1 | 0.501 | 0.480 |
| | BERT | 0.414 | 0.322 | 0.407 | 1 | 0.376 |
| | DeBERTa | 0.348 | 0.343 | 0.348 | 0.351 | 1 |

Table 21: $wMPPC^{source \rightarrow target}$ (last layer) on Flowers-102, for all 10 large models as source, image encoders as target

| | Target / Source | Image | | | | |
|---|---|---|---|---|---|---|
| | | CLIP (I) | SigLIP2 (I) | DFN (I) | DinoV2 | ViT |
| Image | CLIP (I) | 1 | 0.418 | 0.410 | 0.317 | 0.335 |
| | SigLIP2 (I) | 0.413 | 1 | 0.434 | 0.383 | 0.370 |
| | DFN (I) | 0.408 | 0.436 | 1 | 0.350 | 0.359 |
| | DinoV2 | 0.328 | 0.398 | 0.371 | 1 | 0.366 |
| | ViT | 0.332 | 0.377 | 0.358 | 0.369 | 1 |
| Text | CLIP (T) | 0.238 | 0.287 | 0.260 | 0.175 | 0.239 |
| | SigLIP2 (T) | 0.079 | 0.087 | 0.085 | 0.097 | 0.080 |
| | DFN (T) | 0.271 | 0.326 | 0.298 | 0.195 | 0.272 |
| | BERT | 0.115 | 0.131 | 0.122 | 0.126 | 0.122 |
| | DeBERTa | 0.105 | 0.111 | 0.114 | 0.113 | 0.113 |

Table 22: $wMPPC^{source \rightarrow target}$ (last layer) on Flowers-102, for all 10 large models as source, text encoders as target

| | Target / Source | Text | | | | |
|---|---|---|---|---|---|---|
| | | CLIP (T) | SigLIP2 (T) | DFN (T) | BERT | DeBERTa |
| Image | CLIP (I) | 0.206 | 0.067 | 0.221 | 0.093 | 0.094 |
| | SigLIP2 (I) | 0.209 | 0.070 | 0.223 | 0.097 | 0.099 |
| | DFN (I) | 0.223 | 0.070 | 0.238 | 0.099 | 0.102 |
| | DinoV2 | 0.150 | 0.067 | 0.160 | 0.086 | 0.084 |
| | ViT | 0.202 | 0.066 | 0.216 | 0.092 | 0.094 |
| Text | CLIP (T) | 1 | 0.209 | 0.552 | 0.269 | 0.280 |
| | SigLIP2 (T) | 0.329 | 1 | 0.216 | 0.302 | 0.339 |
| | DFN (T) | 0.582 | 0.201 | 1 | 0.273 | 0.281 |
| | BERT | 0.224 | 0.214 | 0.203 | 1 | 0.240 |
| | DeBERTa | 0.227 | 0.244 | 0.198 | 0.231 | 1 |

Table 23: $MPPC^{source \rightarrow target}$ (all layers) on Flowers, image encoders as target

| Source \ Target | | Image CLIP (I) | Image SigLIP2 (I) | Image DinoV2 |
|---|---|---|---|---|
| Image | CLIP (I) | 1 | 0.313 | 0.342 |
| | SigLIP2 (I) | 0.612 | 1 | 0.595 |
| | DinoV2 | 0.373 | 0.355 | 1 |
| Text | CLIP (T) | 0.121 | 0.086 | 0.124 |
| | SigLIP2 (T) | 0.215 | 0.081 | 0.199 |
| | BERT | 0.164 | 0.081 | 0.161 |

Table 24: $MPPC^{source \rightarrow target}$ (all layers) on Flowers, text encoders as target

| Source \ Target | | Text CLIP (T) | Text SigLIP2 (T) | Text BERT |
|---|---|---|---|---|
| Image | CLIP (I) | 0.142 | 0.118 | 0.125 |
| | SigLIP2 (I) | 0.185 | 0.136 | 0.155 |
| | DinoV2 | 0.168 | 0.138 | 0.148 |
| Text | CLIP (T) | 1 | 0.416 | 0.454 |
| | SigLIP2 (T) | 0.309 | 1 | 0.355 |
| | BERT | 0.393 | 0.401 | 1 |

Table 25: $MPPC^{source \rightarrow target}$ (last layer) on Flowers, image encoders as target

| Source \ Target | | Image CLIP (I) | Image SigLIP2 (I) | Image DinoV2 |
|---|---|---|---|---|
| Image | CLIP (I) | 1 | 0.356 | 0.309 |
| | SigLIP2 (I) | 0.387 | 1 | 0.386 |
| | DinoV2 | 0.315 | 0.377 | 1 |
| Text | CLIP (T) | 0.113 | 0.083 | 0.078 |
| | SigLIP2 (T) | 0.072 | 0.054 | 0.081 |
| | BERT | 0.098 | 0.063 | 0.063 |

Table 26: $MPPC^{source \rightarrow target}$ (last layer) on Flowers, text encoders as target

| Source \ Target | | Text CLIP (T) | Text SigLIP2 (T) | Text BERT |
|---|---|---|---|---|
| Image | CLIP (I) | 0.179 | 0.128 | 0.129 |
| | SigLIP2 (I) | 0.171 | 0.121 | 0.124 |
| | DinoV2 | 0.155 | 0.112 | 0.115 |
| Text | CLIP (T) | 1 | 0.325 | 0.341 |
| | SigLIP2 (T) | 0.138 | 1 | 0.180 |
| | BERT | 0.448 | 0.242 | 1 |

Table 27: Main libraries and code used in the paper

| Library | Source (URL) | Licence (URL) |
|---|---|---|
| PyTorch | https://github.com/pytorch/pytorch | BSD |
| Numpy | https://github.com/numpy/numpy | BSD |
| OpenCLIP | https://github.com/mlfoundations/open_clip | MIT |
| HF Transformers | https://github.com/huggingface/transformers | Apache 2.0 |
| HF Datasets | https://github.com/huggingface/datasets | Apache 2.0 |

