# OpenReview forum: "Explaining How Visual, Textual and Multimodal Encoders Share Concepts"
_ICLR.cc/2026/Conference — Submitted to ICLR 2026_

### Official Review · Reviewer_7ViG · 2025-10-29

**Soundness:** 2
**Presentation:** 3
**Contribution:** 2
**Rating:** 2
**Confidence:** 4

**Summary:**

This paper presents a comparison of visual, textual and multimodal encoders based on Sparse Auto Encoder (SAE) derived features. It proposes two tools - weighted Maximum Pairwise Pearson Correlation and Comparative Sharedness for performing this comparison. Using these tools, the model examines three datasets of text-image pairs (COCO, Laion-2B and Oxford-102 flowers) and reports some findings. The key findings are that 1) Encoders from different modalities (visual and text) show greater similarities when considering representations from only the final layer, 2) cross-modal similarities appear to be correlated to the image-text alignment quality in the dataset, and 3) visual features specific to visual language models such as CLIP/DFN are more similar to those from LLMs (e.g. BERT) than from other visual foundation models (e.g. Dino v2).

**Strengths:**

* Proposes two new tools for comparing visual and text language models based on sparse auto-encoder (SAE) derived features
* Presents a comparison of multiple vision, text and multimodal encoders based on SAE derived features.
* Obtains some findings about the similarity of models, data quality and representations.

**Weaknesses:**

* The findings in the paper (e.g. importance of the last layer, the typology of features specific to visual language models are more similar to text language models rather than other visual foundation models) could have been obtained due to idiosyncrasies in the specific datasets and need to be confirmed by repeating the analysis on more datasets. Without such a confirmation, it is hard to know whether these findings will generalize to novel datasets and models.
* The paper employs SAE but does not present an overview of the SAE approach. Without such a presentation, it is hard to understand the significance of specific modifications that were employed in the paper (e.g. in Section 2).

**Questions:**

* Section 2: It would be good to present a short overview of SAE in the appendix. Without such an overview, it is hard to comprehend sentences like L062: "The SAE is trained with mean square error loss using all patches or token of the input text"
* L125: images -> examples
* L374: "The obtained typology is very similar to the one established while considering VLM visual encoders, pushing the hypothesis that previous observations could be caused by their text pretraining." Could there be other factors that could have resulted in this finding?

---

> ### Author Response · Authors · 2025-11-21
>
> We thank the reviewer for their time and constructive feedback. We appreciate the recognition of our proposed tools and experimental setup. Below, we address the reviewer’s concerns regarding dataset coverage, methodological clarity, and interpretation of findings, and provide additional explanations where necessary.
>
> ### Experimental context
> Previous comparable studies [1] [2] usually consider 1 dataset and two to three models. In opposition, we analyze twenty one encoders of different types and sizes, on 3 different text-image datasets. To ensure significance of our findings, we perform a large-scale systematic study, with results to be found in tables 1-3 and 7-22. We also analyze the statistical significance of MPPC and wMPPC in Appendix C.
>
> ### Presentation of Sparse Autoencoders
> Sparse Autoencoders are presented in section 2 (lines 56-60) and they very common and known approaches in interpretability papers. In that field the first paper is from 2023 and is cited in our manuscript (line 29). We also report all the main approaches dealing with various types of SAEs in the related works (418-425 for SAE). It is true that the "related works" are sometimes presented in Section 2 but we chose to adopt a SOTA reporting after the work to ease the discussion. Such a choice has been made in many previous ICLR papers.
>
> ### Typographical error
> We have addressed and corrected the typographical error on L125 highlighted by the reviewer to enhance clarity.
>
> ### Possible cause of Comparative Sharedness findings
> > "The obtained typology is very similar to the one established while considering VLM visual encoders, pushing the hypothesis that previous observations could be caused by their text pretraining." Could there be other factors that could have resulted in this finding?
>
> We use Generalized Comparative Sharedness with grouped of multiple models trained on general purpose datasets in order to make our findings more significant. Such behaviour could in theory have connections with other factors than their pretraining, but it still seems to be the more logical explaination.
>
> [1] Wang, Junxuan, et al. "Towards Universality: Studying Mechanistic Similarity Across Language Model Architectures." The Thirteenth International Conference on Learning Representations.
>
> [2] Thasarathan, Harrish, et al. "Universal sparse autoencoders: Interpretable cross-model concept alignment." Forty-second International Conference on Machine Learning. 2025.

---

> > ### Comment · Reviewer_7ViG · 2025-11-25
> >
> > Thanks for your responses.
> >
> > My concerns are not addressed by the responses.
> > * As I mentioned earlier, the findings in the paper could have been obtained due to peculiarities in the datasets and thus need to be corroborated by reporting results on additional datasets. Thus I am not convinced about the generalizability of these findings.
> > * Given that the paper makes use of SAE, it would be good to present a concise overview of SAE to make it easier to follow for the readers.

---

> > > ### Author Response · Authors · 2025-11-25
> > > **Both points are addressed**
> > >
> > > * SAE **are** presented in the paper and **we cite all the main papers** that use SAE in the same perspective as us
> > >     - in **section 4**:(Bricken et al., 2023; Cunningham et al., 2023).   (Lieberum et al., 2024; Gao et al., 2025; Rajamanoharanet al., 2024).  (Gorton,2024; Thasarathan et al., 2025; Lim et al., 2025; Rao et al., 2024; Papadimitriou et al., 2025; Pachet al., 2025; Yan et al., 2025; Zaigrajew et al.)   (Lim et al.,2025).   (Fel et al., 2023)
> > >   - Which papers are omitted ?
> > > * Our experiments are conducted on **three datasets**. The results are consistent over the **21 encoders** tested.
> > >   - it would be surprising that to get "pecularities" over three datasets and so many models
> > >   - beyond using two generic dataset, it is the first time a paper report results on a specific domain
> > >   - papers in the domain  usually test on one dataset only
> > >   - there is still a limited place to report results
> > >   - we provide the code to test on other datasets
> > >
> > > Would it be possible to be more *specific* on what is exactly expected beyond  what is already reported ? In particular **which dataset** would be relevant and why is it different from the datasets we use and those used in previous papers ?

---

### Official Review · Reviewer_muUv · 2025-11-01

**Soundness:** 2
**Presentation:** 3
**Contribution:** 2
**Rating:** 4
**Confidence:** 2

**Summary:**

The authors propose a novel indicator to compare multiple encoders using their SAE features. This allows the quantification of the “shared” features common to different models. The key contributions are the scale of comparisons and performing multimodal comparisons. The authors also propose 2 metrics – weighted Maximum Pairwise  Pearson  Correlation and Comparative Sharedness which enable comparison across models.

**Strengths:**

1.	Extensive experiments have been conducted – with/outside modality comparisons in popular models, multiple datasets
2.	Qualitative analysis to identify underlying concepts has been done

**Weaknesses:**

My main concern is that all the numbers reported are in terms of the proposed metric wMPPC. While I understand intuitively why the use of weighting is important, I think there needs to be a comparison between wMPPC and older metrics, with examples that show the need for proposing wMPPC.

Detailed questions and clarifications are listed in the Questions section.

**Questions:**

1.	In the introduction, please elaborate why handling multiple modalities is challenging, and why this has not been done in the past.
2.	The claim in the abstract that previous papers have only looked at single modalities may need more context. Other papers like [1-4] do appear to have considered multi modality. Please clarify why these were not included.
3.	In equation (6), can we not just use the term $S_i^M \times (\rho_i^{M -> A} - \rho_i^{M-> B})$? This term would decrease if the similarity between M and B increases, and increase if the similarity between M and A increase. I’m guessing it’s to handle the positive and negative values of $\rho$ but it would be good to clarify this.
4.	Likewise, in eq (7), please provide the intuition for choosing this particular form, as opposed to taking the absolute value of the difference.
5.	I may have missed it, but please highlight the area where models of different sizes have been experimented with.
6.	Could you also highlight all the 21 models mentioned in the abstract in the appendix?
7.	In the implementation details, please describe how the hyperparameter tuning was done.
8.	The conclusion about the last layers of the LLMs being most important semantically is drawn from Figure 1, which is based on the proposed metric. Is this conclusion also supported with MPPC metric as well?
9.	I do not fully understand how the conclusion made about the last layers being important holds, given that in Fig1, towards the inner layers (~20), the wMPPC values are extremely low across the layers.
10.	My main concern is that all the numbers reported are in terms of the proposed metric wMPPC. While I understand intuitively why the use of weighting is important, I think there needs to be a comparison between wMPPC and older metrics, with examples that show the need for proposing wMPPC.


[1] Isabel Papadimitriou, Huangyuan Su, Thomas Fel, Sham Kakade, and Stephanie Gil. Interpreting the linear structure of vision-language model embedding spaces. arXiv preprint arXiv:2504.11695, 2025.

[2] Mateusz Pach, Shyamgopal Karthik, Quentin Bouniot, Serge Belongie, and Zeynep Akata. Sparse autoencoders learn monosemantic features in vision-language models. arXiv preprint arXiv:2504.02821, 2025.

[3] Hanqi Yan, Xiangxiang Cui, Lu Yin, Paul Pu Liang, Yulan He, and Yifei Wang. Multi-faceted multimodal monosemanticity. arXiv preprint arXiv:2502.14888, 2025.

[4] Vladimir Zaigrajew, Hubert Baniecki, and Przemyslaw Biecek. Interpreting CLIP with hierarchical sparse autoencoders. arXiv preprint arXiv:2502.20578, 2025.

---

> ### Author Response · Authors · 2025-11-21
>
> We thank the reviewer for their constructive assessment and for recognizing the breadth of experiments and qualitative analyses in our work. We appreciate the thoughtful questions regarding the motivation, mathematical formulation, and comparative validation of our proposed metrics.
>
> ### Positioning
> Previous works indeed deal with multimodal interpretability. However, those works usually focus on specific models comprising both image and text encoders [1-4], in order to extract multimodal features and without allowing quantitative comparison with models outside this paradigm (such as DinoV2 or Deberta). We included the [1-4] as references for completeness, and a remark in the introduction to precise our positioning.
>
>
> ### Comparative Sharedness Formulation (Eq.6-7)
> The suggested formulation has been tested beforehand, but the results were less significant. The reason was that in such a case the Comparative sharedness mainly depended on $\rho^{M\rightarrow A}$ and thus led to less diversity. We added a note on this point in the text. The Generalized Comparative Sharedness (Eq.7) is a generalization to groups of multiple models, therefore it has the same justification.
>
> ### Different sizes
> Experiments have be conducted with both *base* and *large* sized models. Results are present in Table 3 and Tables 11-14.
>
> ### Details on the used models
> All the 21 encoders we used are listed in Appendix A. We clarified Table 4 regarding VLMs.
>
> ### Hyperparameter tuning
> Section 3.1. details the hyperparameter tuning. The TopK sparsity is fixed to 32 as being the smallest powerof 2 obtaining no dead latents on COCO with CLIP-ViT-L/14. We use an expansion factor of 8 as it is used by previous works using TopKSAEs to compare CLIP, SigLIP and DinoV2 [7].
>
> ### Last layer being most important
> We claim that most *shared* information lives within the last layers. Figure 1 heatmaps have a heavy diagonal pattern, highlighting that the last layer (holding high-level semantic information in encoder models) shared only a small amount of information with earlier layers. Similar observation have been made by previous works [5], including MPPC analysis [6].
>
> ### Metric Comparison
> For completeness, we added results with the MPPC metric in Appendix E (tables 23 to 26)
>
> [1] Papadimitriou, Isabel, et al. "Interpreting the linear structure of vision-language model embedding spaces." arXiv preprint arXiv:2504.11695 (2025).
>
> [2] Pach, Mateusz, et al. "Sparse autoencoders learn monosemantic features in vision-language models." arXiv preprint arXiv:2504.02821 (2025).
>
> [3] Yan, Hanqi, et al. "Multi-Faceted Multimodal Monosemanticity." arXiv preprint arXiv:2502.14888 (2025).
>
> [4] Zaigrajew, Vladimir, Hubert Baniecki, and Przemyslaw Biecek. "Interpreting CLIP with Hierarchical Sparse Autoencoders." Forty-second International Conference on Machine Learning.
>
> [5] Wang, Junxuan, et al. "Towards Universality: Studying Mechanistic Similarity Across Language Model Architectures." The Thirteenth International Conference on Learning Representations.
>
> [6] Fel, Thomas, et al. "A holistic approach to unifying automatic concept extraction and concept importance estimation." Advances in Neural Information Processing Systems 36 (2023): 54805-54818.
>
> [7] Thasarathan, Harrish, et al. "Universal sparse autoencoders: Interpretable cross-model concept alignment." Forty-second International Conference on Machine Learning. 2025.

---

### Official Review · Reviewer_WLJ6 · 2025-11-01

**Soundness:** 3
**Presentation:** 2
**Contribution:** 2
**Rating:** 2
**Confidence:** 4

**Summary:**

The paper introduces two new metrics: weighted Maximum Pairwise Pearson Correlation (wMPPC), which is an extension of the previous MPPC score, and Comparative Sharedness, to compare interpretable features extracted via Sparse Autoencoders (SAEs) across visual, textual, and multimodal encoders. The authors conduct a large-scale analysis on 21 transformer-based encoders from different modalities and datasets, identifying shared and modality-specific concepts. Results highlight that shared cross-modal information mainly resides in the final layers and that text pretraining drives high-level visual concepts in VLMs.

**Strengths:**

* Analyzing similarities and differences across visual, textual, and multimodal encoders is valuable, as it can inform how future models are trained and aligned

* The study spans a large and diverse set of 21 transformer encoders, offering broad coverage across modalities, datasets, and scales.

* The paper includes a detailed limitation section, showing good awareness of scope boundaries and possible extensions.

**Weaknesses:**

* I found a bit surprising that CLIP image features are more correlated with DINOv2 image or than with SigLIP image (trained similarly to CLIP), Tab. 1. Same for SigLIP image being more correlated to CLIP and BERT text rather than SigLIP text encoder ! This make me question the proposed metrics.

* It’s not clear how the observed correlations translate to real-world impact (measured with quantitative metrics), for example, whether they relate to model performance, bias, or hallucination behavior.

* The study focus on contrastive multimodal encoders. How these findings holds for encoders trained with reconstruction objectives such as AIMv2 [1]

* The analysis and findings (e.g. increasing correlation in last layers, multimodal concepts ...) closely resembles earlier concept-based interpretability [2] and modality alignment papers [3], but the connection to those frameworks is not mentioned or clarified. The paper should position itself to these related lines of research.

* For this kind of papers, more visual illustrations might help to understand better the contributions.

[1] "Multimodal autoregressive pre-training of large vision encoders", CVPR 2025.

[2] "A concept-based explainability framework for large multimodal models." NeurIPS 2024.

[3] "Implicit multimodal alignment: On the generalization of frozen llms to multimodal inputs." NeurIPS 2024.

**Questions:**

Please check weaknesses section.

---

> ### Author Response · Authors · 2025-11-21
>
> We thank the reviewer for their careful reading and constructive feedback. We appreciate the recognition of our work’s large-scale scope, methodological clarity, and its potential relevance for understanding multimodal encoders.
>
> ### Suprising results
> First, note that results on Table 2 are more in line with the usual intuition. This last deals with the last layer that can be considered as more significant as pointed on lines 259-260. These results in Table 2 are in line with previous studies (cf. lines 254-255) thus making the proposed wMPPC relevant. The fact that Table 1 reports "surprising results" should thus be considered as interesting since it exhibits novels results that goes beyond the usual belief in the field.
>
> ### Real-world impact
> We propose a method to quantitatively compare different models, upon their inner, interpretable concepts. Therefore, we do not evaluate the performance of those frozen networks. However, previous studies [1] suggest that networks' representations get more similar as performance increases, and further research could be conducted in this direction. Also, we focus on comparing a wide variety of 21 models, spanning text, vision and multimodal encoders, that is already significantly more that previous comparable study [2] [3] [4].
>
> ### Positioning
> Our work focuses on quantitative comparison of different networks. However [2] [3] are indeed relevant works in the field of multimodal interpretability, as they deal with multimodal dictionary learning, and multimodal alignment. We included them as references.
>
> [1] Huh, Minyoung, et al. "Position: The platonic representation hypothesis." Forty-first International Conference on Machine Learning. 2024.
>
> [2] Parekh, Jayneel, et al. "A concept-based explainability framework for large multimodal models." Advances in Neural Information Processing Systems 37 (2024): 135783-135818.
>
> [3] Shukor, Mustafa, and Matthieu Cord. "Implicit multimodal alignment: On the generalization of frozen llms to multimodal inputs." Advances in Neural Information Processing Systems 37 (2024): 130848-130886.
>
> [4] Wang, Junxuan, et al. "Towards Universality: Studying Mechanistic Similarity Across Language Model Architectures." The Thirteenth International Conference on Learning Representations.

---

### Official Review · Reviewer_QAEJ · 2025-11-11

**Soundness:** 3
**Presentation:** 3
**Contribution:** 3
**Rating:** 6
**Confidence:** 2

**Summary:**

The paper presents a new metric "weighted Maximum Pairwise Pearson Correlation" or wMPPC which is similarity measure between two models computed through the weighted expectation of per feature correlation by sampling the activations. The weighting allows focus on correlations that are high between a set of features in two models. The authors find that computing wMPPC on different models uncovers differences in the quality of image-text alignment between datasets, e.g. Laion-2B is worse that Coco.  Another metric "Generalized Comparative Sharedness" is proposed that allows probing of a model over individual concepts/features to determine how unique it is to a group/class of models. The former is global metric of model similarity, the latter is more focused metric of similarity. The sharedness metrics shows how some textual concepts are well shared between text and VLMs, but not visual foundation models. These two new metrics show some promise in being useful diagnostics to help understand encoders.

**Strengths:**

Two new metrics are proposed that help to understand the similarities and differences between models. The authors show how these metrics can be used to uncover interesting details like the quality of the original corpora or "shared concepts" learned between models. The paper provides clear details and pointers to scripts on reproducing results and works with public data sets and models so should be highly reproducible.

**Weaknesses:**

The paper provides a comparative study of visual, textual and joint vision-text models. It would be super interesting to see what insights these measures could provide with the addition of audio to the assessed modalities.

**Questions:**

Have you looked at adding audio as a modality to analyze with your two new metrics?

---

> ### Author Response · Authors · 2025-11-21
>
> We thank the reviewer for the thoughtful feedback and for highlighting the potential of extending our analysis to include audio. We agree that incorporating audio modalities could indeed provide additional insights. However, our current study focuses on encoders trained on visual, textual, and multimodal (image–text) data. Extending our proposed metrics, wMPPC and Comparative Sharedness, to include audio would require access to multimodal datasets that jointly contain audio, image, and text modalities. Consequently, we did not conduct audio experiments in this work. Nonetheless, the proposed framework is theoretically compatible with audio and other modalities, and we see this as an exciting direction for future work.

---

### Meta-Review · Area_Chair_XLCW · 2026-01-05

**Summary:**

Reviewers primarily doubted the metric's reliability due to counter-intuitive results and questioned the generalizability of the findings across datasets. Despite the broad experimental scale, the paper lacked a clear technical overview of Sparse Autoencoders (SAEs) and failed to convincingly bridge its findings with existing multimodal research.

**Reviewer Concerns:**

The authors successfully addressed concerns regarding hyperparameter tuning, model scale, and the inclusion of baseline MPPC comparisons in the appendix. However, concerns regarding the idiosyncrasies of the datasets and the lack of a concise, self-contained overview of SAEs remain outstanding. Reviewers (particularly 7ViG and WLJ6) were not convinced that the results would hold across more diverse data or that the metric's "surprising" findings were a sign of novelty rather than a flaw in the metric itself.

**Reviewer Scores:**

QAEJ (6): Likely 5; low confidence would defer to the stronger technical critiques of the metric.

WLJ6 (2): Stays 2; the rebuttal failed to explain why unrelated models showed high correlation.

muUv (4): Up to 5; technical requests for baselines and size details were satisfied.

7ViG (2): Stays 2; explicitly stated that concerns on dataset variety and SAE clarity were not met.

---

### Decision · Program_Chairs · 2026-01-26

Reject